# Deep neural network models reveal interplay of peripheral coding and stimulus statistics in pitch perception

Mark R. Saddler[1,2,3,5]✉, Ray Gonzalez[1,2,3,5] & Josh H. McDermott [1,2,3,4]✉

Perception is thought to be shaped by the environments for which organisms are optimized. These influences are difficult to test in biological organisms but may be revealed by machine perceptual systems optimized under different conditions. We investigated environmental and physiological influences on pitch perception, whose properties are commonly linked to peripheral neural coding limits. We first trained artificial neural networks to estimate fundamental frequency from biologically faithful cochlear representations of natural sounds. The best-performing networks replicated many characteristics of human pitch judgments. To probe the origins of these characteristics, we then optimized networks given altered cochleae or sound statistics. Human-like behavior emerged only when cochleae had high temporal fidelity and when models were optimized for naturalistic sounds. The results suggest pitch perception is critically shaped by the constraints of natural environments in addition to those of the cochlea, illustrating the use of artificial neural networks to reveal underpinnings of behavior.

[1] Department of Brain and Cognitive Sciences, MIT, Cambridge, MA, USA. [2] McGovern Institute for Brain Research, MIT, Cambridge, MA, USA. [3] Center for Brains, Minds and Machines, MIT, Cambridge, MA, USA. [4] Program in Speech and Hearing Biosciences and Technology, Harvard University, Cambridge, MA, USA. [5]These authors contributed equally: Mark R. Saddler, Ray Gonzalez. ✉email: msaddler@mit.edu; jhm@mit.edu

A key goal of perceptual science is to understand why sensory-driven behavior takes the form that it does. In some cases, it is natural to relate behavior to physiology, and in particular to the constraints imposed by sensory transduction. For instance, color discrimination is limited by the number of cone types in the retina[1]. Olfactory discrimination is similarly constrained by the receptor classes in the nose[2]. In other cases, behavior can be related to properties of environmental stimulation that are largely divorced from the constraints of peripheral transduction. For example, face recognition in humans is much better for upright faces, presumably because we predominantly encounter upright faces in our environment[3].

Understanding how physiological and environmental factors shape behavior is important both for fundamental scientific understanding and for practical applications such as sensory prostheses, the engineering of which might benefit from knowing how sensory encoding constrains behavior. Yet, the constraints on behavior are often difficult to pin down. For instance, the auditory periphery encodes sound with exquisite temporal fidelity[4], but the role of this information in hearing remains controversial[5–7]. Part of the challenge is that the requisite experiments—altering sensory receptors or environmental conditions during evolution or development, for instance—are practically difficult (and ethically unacceptable in humans).

The constraints on behavior can sometimes instead be revealed by computational models. Ideal observer models, which optimally perform perceptual tasks given particular sensory inputs and sensory receptor responses, have been the method of choice for investigating such constraints[8]. While biological perceptual systems likely never reach optimal performance, in some cases humans share behavioral characteristics of ideal observers, suggesting that those behaviors are consequences of having been optimized under particular biological or environmental constraints[9–12]. Ideal observers provide a powerful framework for normative analysis, but for many real-world tasks, deriving provably optimal solutions is analytically intractable. The relevant sensory transduction properties are often prohibitively complicated, and the task-relevant parameters of natural stimuli and environments are difficult to specify mathematically. An attractive alternative might be to collect many real-world stimuli and optimize a model to perform the task on these stimuli. Even if not fully optimal, such models might reveal consequences of optimization under constraints that could provide insights into behavior.

In this paper, we explore whether contemporary "deep" artificial neural networks (DNNs) can be used in this way to gain normative insights about complex perceptual tasks. DNNs provide general-purpose architectures that can be optimized to perform challenging real-world tasks[13]. While DNNs are unlikely to fully achieve optimal performance, they might reveal the effects of optimizing a system under particular constraints[14,15]. Previous work has documented similarities between human and network behavior for neural networks trained on vision or hearing tasks[16–18]. However, we know little about the extent to which human-DNN similarities depend on either biological constraints that are built into the model architecture or the sensory signals for which the models are optimized. By manipulating the properties of simulated sensory transduction processes and the stimuli on which the DNN is trained, we hoped to get insight into the origins of behaviors of interest.

Here, we test this approach in the domain of pitch—traditionally conceived as the perceptual correlate of a sound's fundamental frequency (F0)[19]. Pitch is believed to enable a wide range of auditory-driven behaviors, such as voice and melody recognition[20], and has been the subject of a long history of work in psychology[21–25] and neuroscience[26–29]. Yet despite a wealth of data,

the underlying computations and constraints that determine pitch perception remain debated[19]. In particular, controversy persists over the role of spike timing in the auditory nerve, for which a physiological extraction mechanism has remained elusive[30,31]. The role of cochlear frequency selectivity, which has also been proposed to constrain pitch discrimination, remains similarly debated[26,32]. By contrast, little attention has been given to the possibility that pitch perception might instead or additionally be shaped by the constraints of estimating the F0 of natural sounds in natural environments.

One factor limiting resolution of these debates is that previous models of pitch have generally not attained quantitatively accurate matches to human behavior[25,33–40]. Moreover, because most previous models have been mechanistic rather than normative, they do not speak to the potential adaptation of pitch perception to particular types of sounds or peripheral neural codes. Here we used DNNs in the role traditionally occupied by ideal observers, optimizing them to extract pitch information from peripheral neural representations of natural sounds. DNNs have become the method of choice for pitch tracking in engineering applications[41], but have not been combined with realistic models of the peripheral auditory system, and have not been compared to human perception. We then tested the influence of peripheral auditory physiology and natural sound statistics on human pitch perception by manipulating them during model optimization. The results provide new evidence for the importance of peripheral phase locking in human pitch perception. However, they also indicate that the properties of pitch perception reflect adaptation to natural sound statistics, in that systems optimized for alternative stimulus statistics deviate substantially from human-like behavior.

## Results

**Training task and stimuli**. We used supervised deep learning to build a model of pitch perception optimized for natural speech and music (Fig. 1a). DNNs were trained to estimate the F0 of short (50 ms) segments of speech and musical instrument recordings, selected to have high periodicity and well-defined F0s. To emulate natural listening conditions, the speech and music clips were embedded in aperiodic background noise taken from YouTube soundtracks. The networks' task was to classify each stimulus into one of 700 F0 classes (log-spaced between 80 Hz and 1000 Hz, bin width = 1/16 semitones = 0.36% F0). We generated a dataset of 2.1 million stimuli. Networks were trained using 80% of this dataset and the remaining 20% was used as a validation set to measure the success of the optimization.

**Peripheral auditory model**. In our primary training condition, we hard-coded the input representation for our networks to be as faithful as possible to known peripheral auditory physiology. We used a detailed phenomenological model of the auditory nerve[42] to simulate peripheral representations of each stimulus (Fig. 1a). The input representations to our networks consisted of 100 simulated auditory nerve fibers. Each stimulus was represented as a 100-fiber by 1000-timestep array of instantaneous firing rates (sampled at 20 kHz).

An example simulated auditory nerve representation for a harmonic tone is shown in Fig. 1b. Theories of pitch have tended to gravitate toward one of the two axes of such representations: the frequency-to-place mapping along the cochlea's length, or the time axis. However, it is visually apparent that the nerve representation of even this relatively simple sound is quite rich, with a variety of potential cues: phase locking to individual frequencies, phase shifts between these phase-locked responses, peaks in the time-averaged response (the "excitation" pattern) for

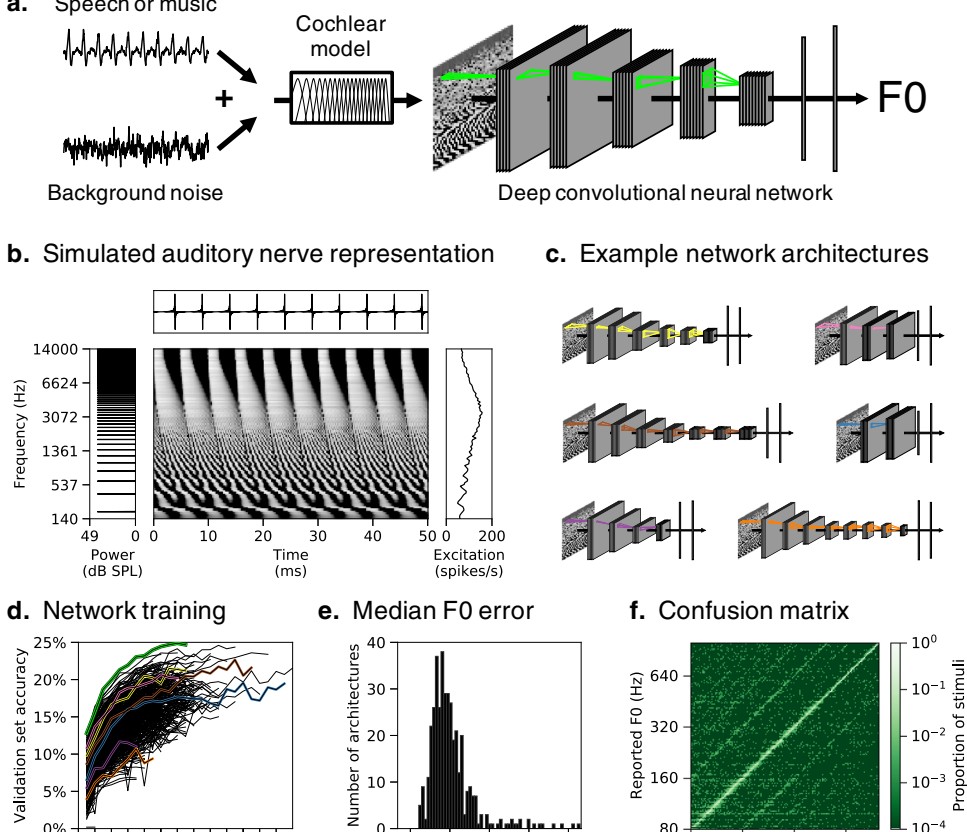

**Fig. 1 Pitch model overview. a** Schematic of model structure. DNNs were trained to estimate the F0 of speech and music sounds embedded in real-world background noise. Networks received simulated auditory nerve representations of acoustic stimuli as input. Green outlines depict the extent of example convolutional filter kernels in time and frequency (horizontal and vertical dimensions, respectively). **b** Simulated auditory nerve representation of a harmonic tone with a fundamental frequency (F0) of 200 Hz. The sound waveform is shown above and its power spectrum is shown to the left. The waveform is periodic in time, with a period of 5ms. The spectrum is harmonic (i.e., containing multiples of the fundamental frequency). Network inputs were arrays of instantaneous auditory nerve firing rates (depicted in greyscale, with lighter hues indicating higher firing rates). Each row plots the firing rate of a frequency-tuned auditory nerve fiber, arranged in order of their place along the cochlea (with low frequencies at the bottom). Individual fibers phase-lock to low-numbered harmonics in the stimulus (lower portion of the nerve representation) or to the combination of high-numbered harmonics (upper portion). Time-averaged responses on the right show the pattern of nerve fiber excitation across the cochlear frequency axis (the "excitation pattern"). Low-numbered harmonics produce distinct peaks in the excitation pattern. **c** Schematics of six example DNN architectures trained to estimate F0. Network architectures varied in the number of layers, the number of units per layer, the extent of pooling between layers, and the size and shape of convolutional filter kernels **d** Summary of network architecture search. F0 classification performance on the validation set (noisy speech and instrument stimuli not seen during training) is shown as a function of training steps for all 400 networks trained. The highlighted curves correspond to the architectures depicted in **a** and **c**. The relatively low overall accuracy reflects the fine-grained F0 bins we used. **e** Histogram of accuracy, expressed as the median F0 error on the validation set, for all trained networks (F0 error in percent is more interpretable than the classification accuracy, the absolute value of which is dependent on the width of the F0 bins). **f** Confusion matrix for the best-performing network (depicted in **a**) tested on the validation set.

low-numbered harmonics, and phase locking to the F0 for the higher-numbered harmonics. The DNN models have access to all of this information. Through optimization for the training task, the DNNs should learn to use whichever peripheral cues best allow them to extract F0.

**Neural network architecture search**. The performance of an artificial neural network is influenced both by the particular weights that are learned during training and by the various parameters that define the architecture of the network[16]. To obtain a high-performing model, we performed a large-scale random architecture search. Each architecture consisted of a feedforward series of layers instantiating linear convolution, nonlinear rectification, normalization, and pooling operations. Within this family, we trained 400 networks varying in their number of layers, number of units per layer, extent of pooling

between layers, and the size and shape of convolutional filters (Fig. 1c).

The different architectures produced a broad distribution of training task performances (Fig. 1d). In absolute terms accuracy was good – the median error was well below 1% (Fig. 1e), which is on par with good human F0 discrimination thresholds[25,43]. The vast majority of misclassifications fell within bins neighboring the true F0 or at an integer number octaves away (Fig. 1f), as in human pitch-matching judgments[44].

**Characteristics of pitch perception emerge in optimized DNNs**. Having obtained a model that can estimate F0 from natural sounds, we simulated a suite of well-known psychophysical experiments to assess whether the model replicated known properties of human pitch perception. Each experiment measures the effect of particular cues on pitch discrimination or estimation

| **Stimulus manipulation and task** | **Human results** | **Model results** |
| --- | --- | --- |

**a.**  Effect of harmonic number and phase on pitch discrimination (*Bernstein & Oxenham, 2005*)

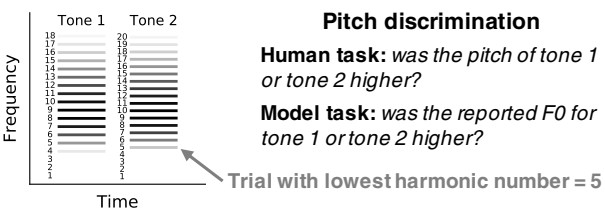

**Pitch discrimination**

**Human task:** *was the pitch of tone 1 or tone 2 higher?*

**Model task:** *was the reported F0 for tone 1 or tone 2 higher?*

**Trial with lowest harmonic number = 5**

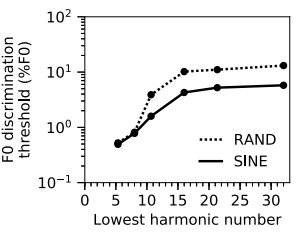
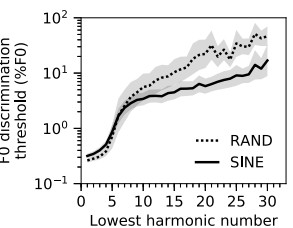

**b.**  Pitch of alternating-phase harmonic complexes (*Shackleton & Carlyon, 1994*)

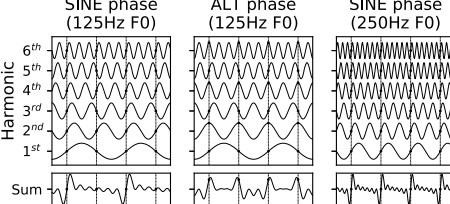

**Pitch estimation**

(detailed in **D**)

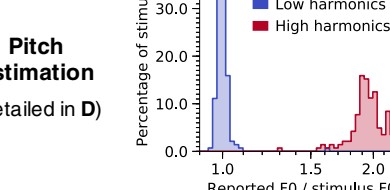
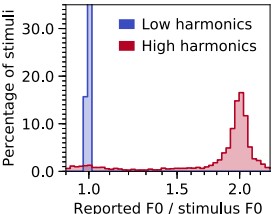

**c.**  Pitch of frequency-shifted complexes (*Moore & Moore, 2003*)

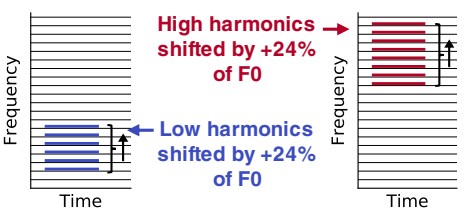

**High harmonics shifted by +24% of F0**

**Low harmonics shifted by +24% of F0**

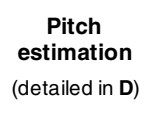

**Pitch estimation**

(detailed in **D**)

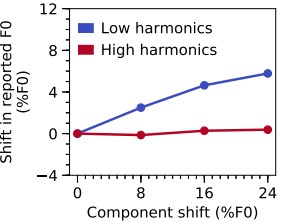
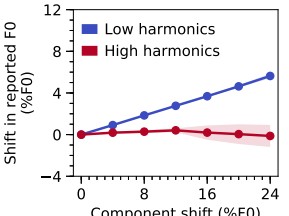

**d.**  Pitch of complexes with individually mistuned harmonics (*Moore et al., 1985*)

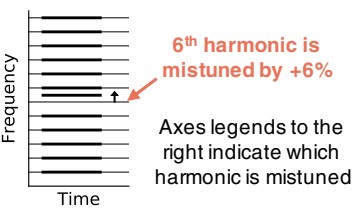

**6th harmonic is mistuned by +6%**

Axes legends to the right indicate which harmonic is mistuned

**Pitch estimation**

**Human task:** *adjust F0 of sine-phase harmonic complex to match pitch of test stimulus*

**Model task:** *report F0 estimate for test stimulus*

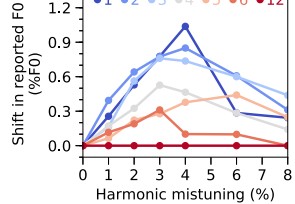
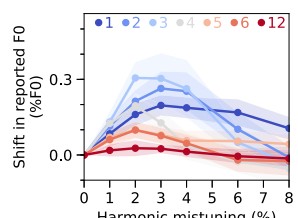

**e.**  Necessity of correct tonotopic representation for pitch discrimination (*Oxenham et al., 2004*)

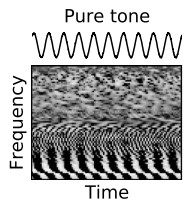
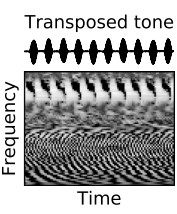

Pure tone

Transposed tone

**Pitch discrimination**

(detailed in **A**)

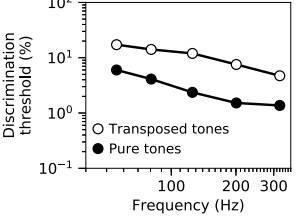
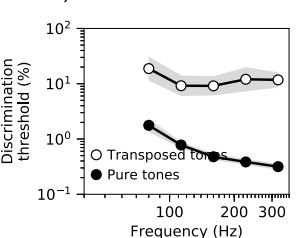

using synthetic tones (Fig. 2, left column), and produces an established result in human listeners (Fig. 2, center column). We tested the effect of these stimulus manipulations on our ten best-performing network architectures. Given evidence for individual differences across different networks optimized for the same task[45], most figures feature results averaged across the ten best networks identified in our architecture search (which we collectively refer to as "the model"). Averaging across an ensemble of networks effectively allows us to marginalize over architectural hyperparameters and provide uncertainty estimates for our

model's results[46,47]. Individual results for the ten networks are shown in Supplementary Fig. 1.

As shown in Fig. 2, the model (right column) qualitatively and in most cases quantitatively replicates the result of each of the five different experiments in humans (center column). We emphasize that none of the stimuli were included in the networks' training set, and that the model was not fit to match human results in any way. These results collectively suggest that the model relies on similar cues as the human pitch system. We describe these results in turn.

**Fig. 2 Pitch model validation: human and neural network psychophysics.** Five classic experiments from the pitch psychoacoustics literature (**a–e**) were simulated on neural networks trained to estimate the F0 of natural sounds. Each row corresponds to a different experiment and contains (from left to right) a schematic of the experimental stimuli, results from human listeners (re-plotted from the original studies), and results from the neural networks. Error bars indicate bootstrapped 95% confidence intervals around the mean of the ten best network architectures when ranked by F0 estimation performance on natural sounds (individual network results are shown in Supplementary Fig. 1). **a** F0 discrimination thresholds for bandpass synthetic tones, as a function of lowest harmonic number and phase. Human listeners and networks discriminated pairs of sine-phase or random-phase harmonic tones with similar F0s. Stimuli were bandpass-filtered to control which harmonics were audible. **b** Perceived pitch of alternating-phase complex tones containing either low or high-numbered harmonics. Alternating-phase tones (i.e., with odd-numbered harmonics in sine phase and even-numbered harmonics in cosine phase) contain twice as many peaks in the waveform envelope as sine-phase tones with the same F0. Human listeners adjusted a sine-phase tone to match the pitch of the alternating-phase tone. Networks made F0 estimates for the alternating-phase tones directly. Histograms show distributions of pitch judgments as the ratio between the reported F0 and the stimulus F0. **c** Pitch of frequency-shifted complexes. Harmonic complexes (containing either low or high-numbered harmonics) were made inharmonic by shifting all component frequencies by the same number of Hz. Human listeners and networks reported the F0s they perceived for these stimuli (same experimental methods as in **b**). Shifts in the perceived F0 are shown as a function of the shift applied to the component frequencies. **d** Pitch of complexes with individually mistuned harmonics. Human listeners and networks reported the F0s they perceived for complex tones in which a single harmonic frequency was shifted (same experimental methods as in **b**). Shifts in the perceived F0 are shown as a function of the mistuning applied to seven different harmonics within the tone (harmonic numbers indicated in different colors at top of graphs). Note that the y-axis limits are different in the human and model graphs—they exhibit qualitative but not quantitative similarity. This could be because the networks are better able to isolate the contribution of the harmonic to the F0, whereas human listeners may sometimes erroneously be biased by the harmonic itself. **e** Frequency discrimination thresholds measured with pure tones and transposed tones. Transposed tones are high-frequency tones that are amplitude-modulated so as to instantiate the temporal cues from low-frequency pure tones at a higher-frequency place on the cochlea. Human and network listeners discriminated pairs of pure tones with similar frequencies and pairs of transposed tones with similar envelope frequencies.

**Dependence on low-numbered harmonics**. First, human pitch discrimination is more accurate for stimuli containing low-numbered harmonics (Fig. 2a, center, solid line)[22,25,43,48]. This finding is often interpreted as evidence for the importance of "place" cues to pitch, which are only present for low-numbered harmonics (Fig. 1b, right). The model reproduced this effect, though the inflection point was somewhat lower than in human listeners: discrimination thresholds were low only for stimuli containing the fifth or lower harmonic (Fig. 2a, right, solid line).

**Phase effects are limited to high-numbered harmonics**. Second, human perception is affected by harmonic phases only for high-numbered harmonics. When harmonic phases are randomized, human discrimination thresholds are elevated for stimuli that lack low-numbered harmonics (Fig. 2a, center, dashed vs. solid line)[25]. In addition, when odd and even harmonics are summed in sine and cosine phase, respectively ("alternating phase", a manipulation that doubles the number of peaks in the waveform's temporal envelope; Fig. 2b, left), listeners report the pitch to be twice as high as the corresponding sine-phase complex, but only for high-numbered harmonics (Fig. 2b, center)[22]. These results are typically thought to indicate use of temporal fluctuations in a sound's envelope when cues for low-numbered harmonics are not available[22,26,43]. The model replicates both effects (Fig. 2a, b, right), indicating that it uses similar temporal cues to pitch as humans, and in similar conditions.

**Pitch shifts for shifted low-numbered harmonics**. Third, frequency-shifted complex tones (in which all of the component frequencies have been shifted by the same number of Hz; Fig. 2c, left) produce linear shifts in the pitch reported by humans, but only if the tones contain low-numbered harmonics (Fig. 2c, center)[23]. The model's F0 predictions for these stimuli resemble those measured from human listeners (Fig. 2c, right).

Fourth, shifting individual harmonics in a complex tone ("mistuning"; Fig. 2d, left) can also produce pitch shifts in humans under certain conditions[21]: the mistuning must be small (effects are largest for 3–4% mistuning) and applied to a low-numbered harmonic (Fig. 2d, center). The model replicates this effect as well, although the size of the shift is smaller than that observed in humans (Fig. 2d, right).

**Poor discrimination of transposed tones**. Fifth, "transposed tones" designed to instantiate the temporal cues from low frequencies at a higher-frequency place on the cochlea (Fig. 2e, left) elicit weak pitch percepts in humans and thus yield higher discrimination thresholds than pure tones (Fig. 2e, center)[24]. This finding is taken to indicate that to the extent that temporal cues to pitch matter perceptually, they must occur at the correct place on the cochlea. The model reproduced this effect: discrimination thresholds were worse for transposed tones than they are for pure tones (Fig. 2e, right).

**DNNs with better F0 estimation show more human-like behavior**. To evaluate whether the human-model similarity evident in Fig. 2 depends on having optimized the model architecture for F0 estimation of natural sounds, we simulated the full suite of psychophysical experiments on each of our 400 trained networks. These 400 networks varied in how well they estimated F0 for the validation set (Fig. 1d, e). For each psychophysical experiment and network, we quantified the similarity between human and network results with a correlation coefficient. We then compared this human-model similarity to each network's performance on the validation set (Fig. 3a–e).

For four of the five experiments (Fig. 3a–d), there was a significant positive correlation between training task performance and human-model similarity ($p < 0.001$ in each case). The transposed tones experiment (Fig. 3e) was the exception, as all networks similarly replicated the main human result regardless of their training task performance. We suspect this is because transposed tones cause patterns of peripheral stimulation that rarely occur for natural sounds. Thus, virtually any model that learns to associate naturally occurring peripheral cues with F0 will exhibit poor performance for transposed tones.

To illustrate the effect of optimization for one experiment, Fig. 3f displays the average F0 discrimination thresholds for each of the worst, middle, and best 10% of networks (sorted by performance on the validation set). It is visually apparent that top-performing networks exhibit more similar psychophysical behavior to humans than worse-performing networks. See Supplementary Fig. 2 for analogous results for the other four experiments from Fig. 2. Overall, these results indicate that networks with better performance on the F0-estimation training task generally exhibit more human-like pitch behavior, consistent

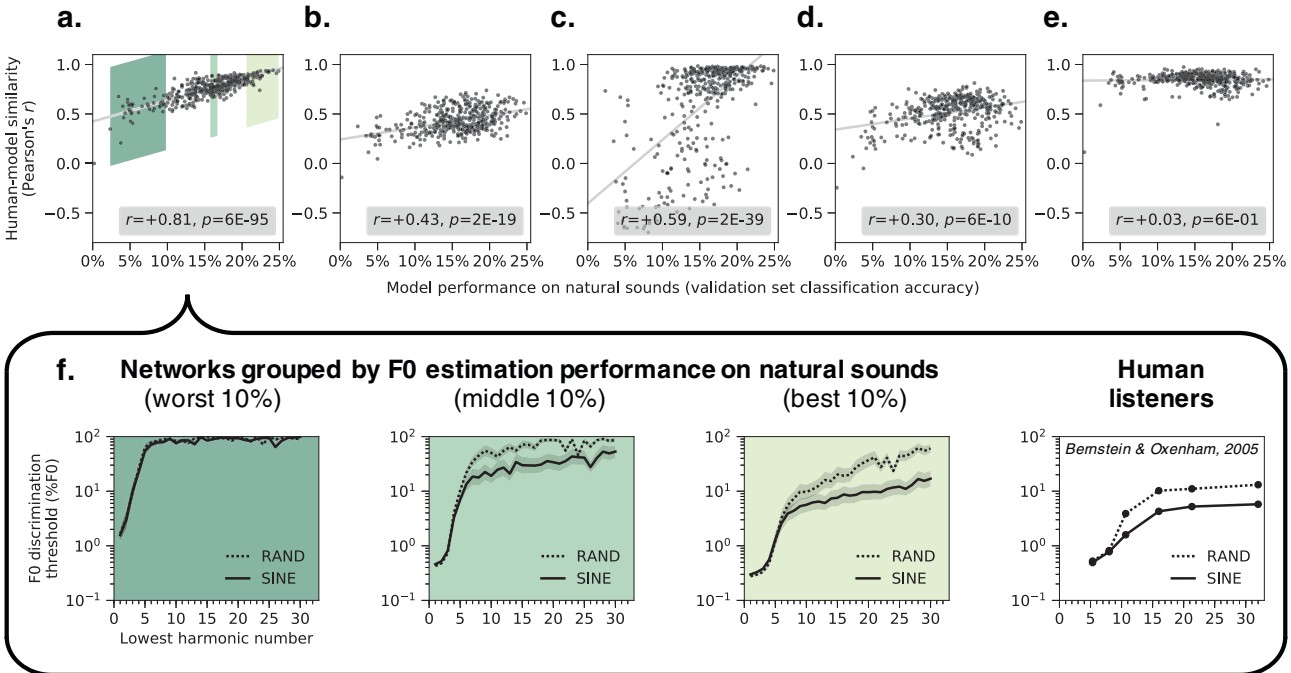

**Fig. 3 Network architectures producing better F0 estimation for natural sounds exhibit more human-like pitch behavior. a–e** Plot human-model similarity in each experiment for all 400 architectures as a function of the accuracy of the trained architecture on the validation set (a set of stimuli distinct from the training dataset, but generated with the same procedure). The similarity between human and model results was quantified for each experiment as the correlation coefficient between analogous data points (Methods). Pearson correlations between validation set accuracy and human-model similarity for each experiment are noted in the legends. Each graph **a–e** corresponds to one of the five main psychophysical experiments (Fig. 2a–e): **a** F0 discrimination as a function of harmonic number and phase, **b** pitch estimation of alternating-phase stimuli, **c** pitch estimation of frequency-shifted complexes, **d** pitch estimation of complexes with individually mistuned harmonics, and **e** frequency discrimination with pure and transposed tones. **f** The results of the experiment from **a** (F0 discrimination thresholds as a function of lowest harmonic number and harmonic phase) measured from the 40 worst, middle, and best architectures ranked by F0 estimation performance on natural sounds (indicated with green patches in **a**). Lines plot means across the 40 networks. Error bars indicate 95% confidence intervals via bootstrapping across the 40 networks. Human F0 discrimination thresholds from the same experiment are re-plotted for comparison.

with the idea that these patterns of behavior are byproducts of optimization under natural constraints.

Because the space of network architectures is large, it is a challenge to definitively associate particular network motifs with good performance and/or human-like behavior. However, we found that very shallow networks both performed poorly on the training task and exhibited less similarity with human behavior (Supplementary Fig. 3). This result provides evidence that deep networks (with multiple hierarchical stages of processing) better account for human pitch behavior than relatively shallow networks.

**Human-like behavior requires a biologically-constrained cochlea.** To test whether a biologically-constrained cochlear model was necessary for human-like pitch behavior, we trained networks to estimate F0 directly from sound waveforms (Fig. 4a). We replaced the cochlear model with a bank of 100 one-dimensional convolutional filters operating directly on the audio. The weights of these first-layer filters were optimized for the F0 estimation task along with the rest of the network.

The learned filters deviated from those in the ear, with best frequencies tending to be lower than those of the hardwired peripheral model (Fig. 4b). Networks with learned cochlear filters also exhibited less human-like behavior than their counterparts with the fixed cochlear model (Fig. 4c, d). In particular, networks with learned cochlear filters showed little ability to extract pitch information from high-numbered harmonics. Discrimination thresholds for higher harmonics were poor (Fig. 4c, Expt. A)

and networks did not exhibit phase effects (Fig. 4c, Expt. A & B). Accordingly, human-model similarity was substantially lower with learned cochlear filters for two of five psychophysical experiments (Fig. 4d; Expt. A: $t(18) = 5.23$, $p < 0.001$, $d = 2.47$; Expt. B: $t(18) = 12.69$, $p < 0.001$, $d = 5.98$). This result suggests that a human-like cochlear representation is necessary to obtain human-like behavior, but also that the F0 estimation task on its own is insufficient to produce a human-like cochlear representation, likely because the cochlea is shaped by many auditory tasks. Thus, the cochlea may be best considered as a constraint on pitch perception rather than the other way around.

**Dependence of pitch behavior on the cochlea.** To gain insight into what aspects of the cochlea underlie the characteristics of pitch perception, we investigated how the model behavior depends on its peripheral input. Decades of research has sought to determine the aspects of peripheral auditory representations that underlie pitch judgments, but experimental research has been limited by the difficulty of manipulating properties of peripheral representations. We took advantage of the ability to perform experiments on the model that are not possible in biology, training networks with peripheral representations that were altered in various ways. To streamline presentation, we present results for a single psychophysical result that was particularly diagnostic: the effect of lowest harmonic number on F0 discrimination thresholds (Fig. 2a, solid line). Results for other experiments are generally congruent with the overall conclusions and are shown in Supplementary Figures. We first present

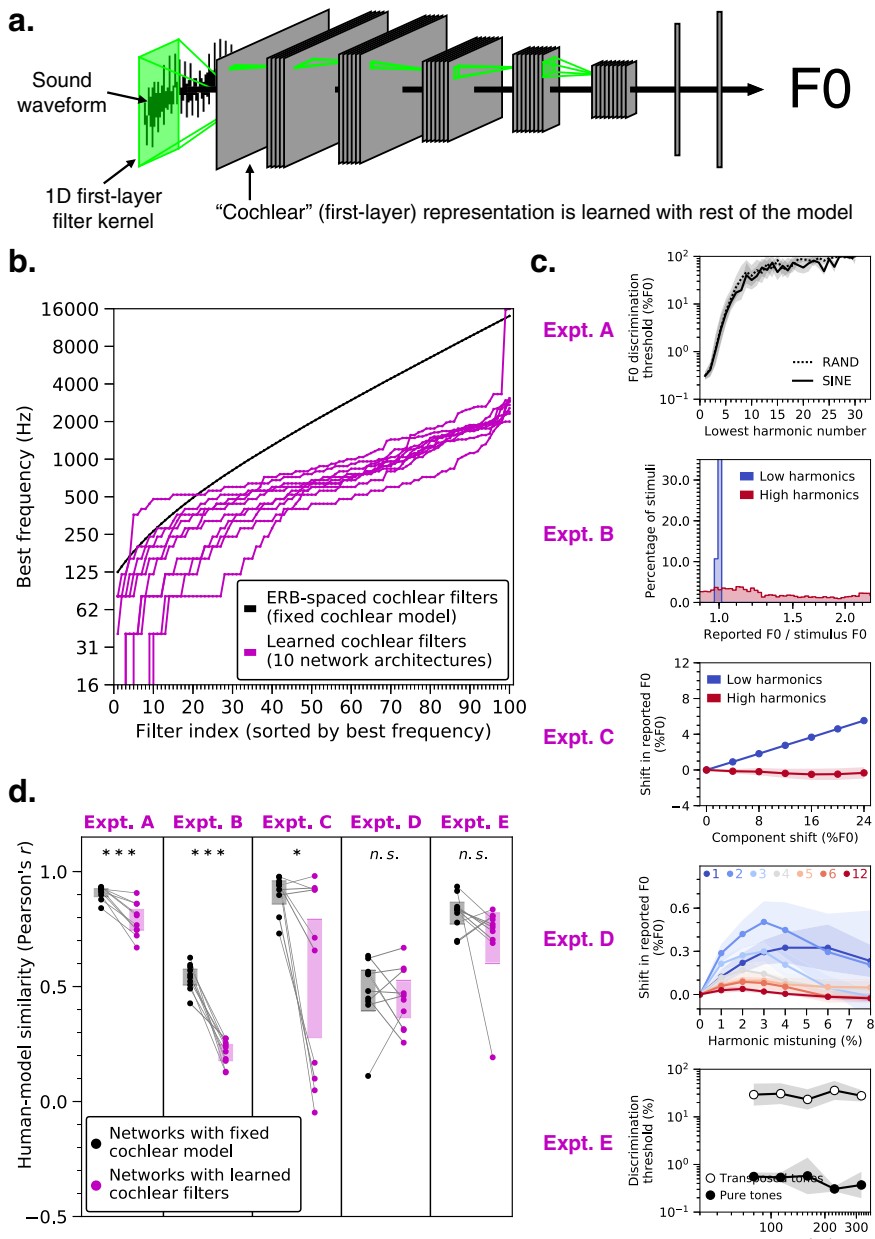

**Fig. 4 Networks trained to estimate F0 directly from sound waveforms exhibit less human-like pitch behavior. a** Schematic of model structure. Model architecture was identical to that depicted in Fig. 1a, except that the hardwired cochlear input representation was replaced by a layer of one-dimensional convolutional filters operating directly on sound waveforms. The first-layer filter kernels were optimized for the F0 estimation task along with the rest of the network weights. We trained the ten best networks from our architecture search with these learnable first-layer filters. **b** The best frequencies (sorted from lowest to highest) of the 100 learned filters for each of the ten network architectures are plotted in magenta. For comparison, the best frequencies of the 100 cochlear filters in the hardwired peripheral model are plotted in black. **c** Effect of learned cochlear filters on network behavior in all five main psychophysical experiments (see Fig. 2a–e): F0 discrimination as a function of harmonic number and phase (Expt. a), pitch estimation of alternating-phase stimuli (Expt. b), pitch estimation of frequency-shifted complexes (Expt. c), pitch estimation of complexes with individually mistuned harmonics (Expt. d), and frequency discrimination with pure and transposed tones (Expt. e). Lines plot means across the ten networks; error bars plot 95% confidence intervals, obtained by bootstrapping across the ten networks. **d** Comparison of human-model similarity metrics between networks trained with either the hardwired cochlear model (black) or the learned cochlear filters (magenta) for each psychophysical experiment. Asterisks indicate statistical significance of two-sample t-tests comparing the two cochlear model conditions: \*\*\*p < 0.001, \*p = 0.016. Error bars indicate 95% confidence intervals bootstrapped across the ten network architectures.

experiments manipulating the fidelity of temporal coding, followed by experiments manipulating frequency selectivity along the cochlea's length.

**Human-like behavior depends critically on phase locking.** To investigate the role of temporal coding in the auditory

periphery, we trained networks with alternative upper limits of auditory nerve phase locking. Phase locking is limited by biophysical properties of inner hair cell transduction[4], which are impractical to alter in vivo but which can be modified in silico via the simulated inner hair cell's lowpass filter[42]. We separately trained networks with lowpass cutoff frequencies of 50 Hz,

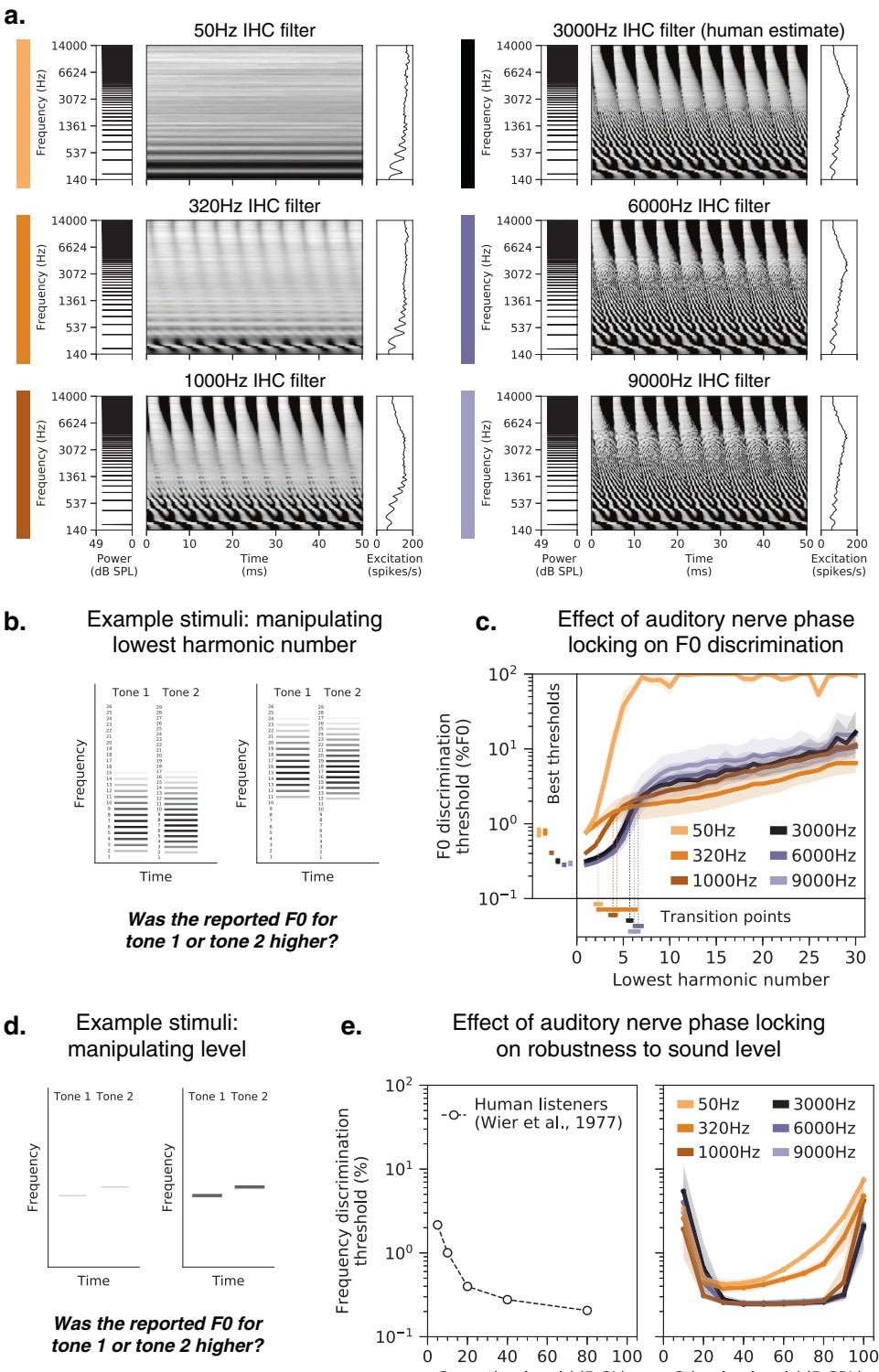

320 Hz, 1000 Hz, 3000 Hz (the nerve model's default value, commonly presumed to roughly match that of the human auditory nerve), 6000 Hz, and 9000 Hz. With a cutoff frequency of 50 Hz, virtually all temporal structure in the peripheral representation of our stimuli was eliminated, meaning the network only had access to cues from the place of excitation along the cochlea (Fig. 5a). As the cutoff frequency was increased, the network gained access to progressively finer-grained spike-timing information (in addition to the place cues). The ten best-performing networks from the architecture search were retrained separately with each of these altered cochleae.

Reducing the upper limit of phase locking qualitatively changed the model's psychophysical behavior and made it less human-like. As shown in Fig. 5b, c, F0 discrimination thresholds became worse, with the best threshold (the left-most data point, corresponding to a lowest harmonic number of 1) increasing as the cutoff was lowered (significantly worse for all three conditions: 1000 Hz, $t(18) = 4.39$, $p < 0.001$, $d = 1.96$; 320 Hz, $t(18) = 11.57$, $p < 0.001$, $d = 5.17$; 50 Hz, $t(18) = 9.30$, $p < 0.001$,

**Fig. 5 Pitch perception is impaired in networks optimized with degraded spike timing in the auditory nerve. a** Simulated auditory nerve representations of the same stimulus (harmonic tone with 200 Hz F0) under six configurations of the peripheral auditory model. Configurations differed in the cutoff frequency of the inner hair cell lowpass filter, which sets the upper limit of auditory nerve phase locking. The 3000 Hz setting is that normally used to model the human auditory system. As in Fig. 1b, each peripheral representation is flanked by the stimulus power spectrum and the time-averaged cochlear excitation pattern. **b** Schematic of stimuli used to measure F0 discrimination thresholds as a function of lowest harmonic number. Gray level denotes amplitude. Two example trials are shown, with two different lowest harmonic numbers. **c** F0 discrimination thresholds as a function of lowest harmonic number measured from networks trained and tested with each of the six peripheral model configurations depicted in **a**. The best thresholds and the transition points from good to poor thresholds (defined as the lowest harmonic number for which thresholds first exceeded 1%) are re-plotted to the left of and below the main axes, respectively. Here and in **e**, lines plot means across the ten networks; error bars plot 95% confidence intervals, obtained by bootstrapping across the ten networks. **d** Schematic of stimuli used to measure frequency discrimination thresholds as a function of sound level. Gray level denotes amplitude. **e** Frequency discrimination thresholds as a function of sound level measured from human listeners (left) and from the same networks as **c** (right). Human thresholds, which are reported as a function of sensation level, are re-plotted from[50].

$d = 4.16$; two-sample t-tests comparing to thresholds in the 3000 Hz condition). This in itself is not surprising, as it has long been known that phase locking enables better frequency discrimination than place information alone[9,49]. However, thresholds also showed a different dependence on harmonic number as the phase locking cutoff was lowered. Specifically, the transition from good to poor thresholds, here defined as the leftmost point where thresholds exceeded 1%, was lower with degraded phase locking. This difference was significant for two of the three conditions (1000 Hz, $t(18) = 5.15$, $p < 0.001$, $d = 2.30$; 50 Hz, $t(18) = 10.10$, $p < 0.001$, $d = 4.52$; two-sample t-tests comparing to the 3000 Hz condition; the transition point was on average lower for the 320 Hz condition, but the results were more variable across architectures, and so the difference was not statistically significant). Increasing the cutoff to 6000 Hz or 9000 Hz had minimal effects on both of these features (Fig. 5c), suggesting that superhuman temporal resolution would not continue to improve pitch perception (at least as assessed here). Discrimination thresholds for high-numbered harmonics were in fact slightly worse for increased cutoff frequencies. One explanation is that increasing the model's access to fine timing information biases the learned strategy to rely more on this information, which is less useful for determining the F0 of stimuli containing only high-numbered harmonics. Overall, these results suggest that auditory nerve phase locking like that believed to be present in the human ear is critical for human-like pitch perception.

A common criticism of place-based pitch models is that they fail to account for the robustness of pitch across sound level, because cochlear excitation patterns saturate at high levels[26]. Consistent with this idea, frequency discrimination thresholds (Fig. 5d) measured from networks with lower phase locking cutoffs were less invariant to level than networks trained with normal spike-timing information (Fig. 5e, right). Thresholds for models with limited phase locking became progressively worse for louder tones, unlike those for humans (Fig. 5e, left)[50]. This effect produced an interaction between the effect of stimulus level and the phase locking cutoff on discrimination thresholds ($F(13.80,149.08) = 4.63$, $p < 0.001$, $\eta^2_{partial} = 0.30$), in addition to the main effect of the cutoff ($F(5,54) = 23.37$, $p < 0.001$, $\eta^2_{partial} = 0.68$; also evident in Fig. 5c). Similar effects were observed when thresholds were measured with complex tones (data not shown).

To control for the possibility that the poor performance of the networks trained with lower phase locking cutoffs might be specific to the relatively small number of simulated auditory nerve fibers in the model, we generated an alternative representation for the 50 Hz cutoff condition, using 1000 nerve fibers and 100 timesteps (sampled at 2 kHz). We then trained and tested the ten best-performing networks from our architecture search on these representations (transposing the nerve fiber and time dimensions to maintain the input size and thus be able to use the same network architecture). Increasing the number of simulated auditory nerve fibers by a full order of magnitude modestly improved thresholds but did not qualitatively change the results: networks without high-fidelity temporal information still exhibited abnormal F0 discrimination behavior. The 50 Hz condition results in Fig. 5c, e are taken from the 1000 nerve fiber networks, as this seemed the most conservative comparison. Results for different numbers of nerve fibers are provided in Supplementary Fig. 4.

We simulated the full suite of psychophysical experiments on all networks with altered cochlear temporal resolution (Supplementary Fig. 5). Several other experimental results were also visibly different from those of humans in models with altered phase locking cutoffs (in particular, the alternating-phase and mistuned harmonics experiments). Overall, the results indicate that normal human pitch perception depends on phase locking up to 3000 Hz.

**Human-like behavior depends less on cochlear filter bandwidths.** The role of cochlear frequency tuning in pitch perception has also been the source of longstanding debates[22,32,43,48,51,52]. Classic "place" theories of pitch postulate that F0 is inferred from the peaks and valleys in the excitation pattern. Contrary to this idea, we found that simply eliminating all excitation pattern cues (by separately re-scaling each frequency channel in the peripheral representation to have the same time-averaged response, without retraining the model) had almost no effect on network behavior (Supplementary Fig. 6). This result suggests that F0 estimation does not require the excitation pattern per se, but it remains possible that it might still be constrained by the frequency tuning of the cochlea.

To investigate the perceptual effects of cochlear frequency tuning, we trained networks with altered tuning. We first scaled cochlear filter bandwidths to be two times narrower and two times broader than those estimated for human listeners[53]. The effect of this manipulation is visually apparent in the width of nerve fiber tuning curves as well as in the number of harmonics that produce distinct peaks in the cochlear excitation patterns (Fig. 6a).

We also modified the cochlear model to be linearly spaced (Fig. 6b), uniformly distributing the characteristic frequencies of the model nerve fibers along the frequency axis and equating their filter bandwidths. Unlike a normal cochlea, which resolves only low-numbered harmonics, the linearly spaced alteration yielded a peripheral representation where all harmonics are equally resolved by the cochlear filters, providing another test of the role of frequency selectivity.

Contrary to the notion that cochlear frequency selectivity strongly constrains pitch discrimination, networks trained with different cochlear bandwidths exhibit relatively similar

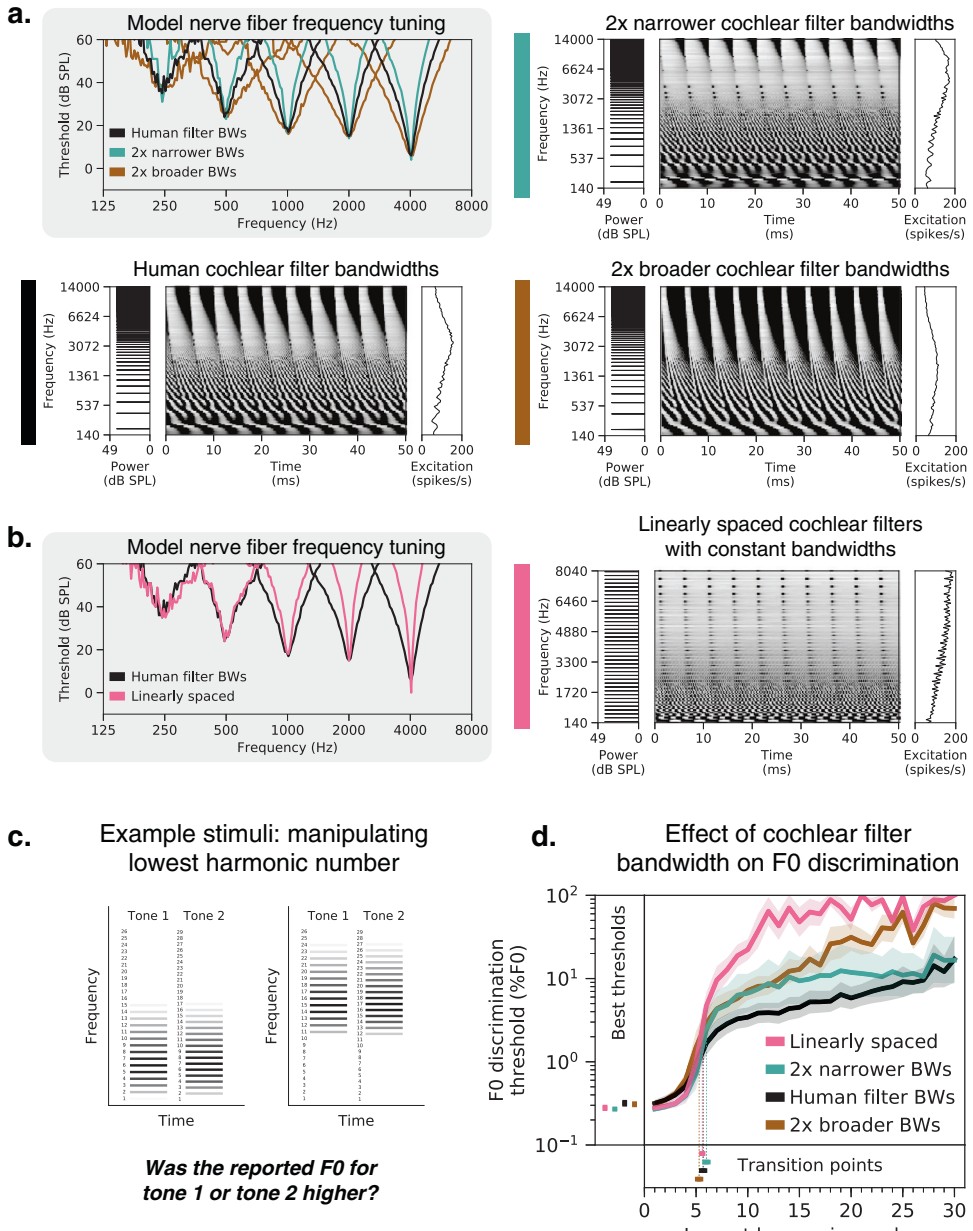

**Fig. 6 Cochlear frequency tuning has relatively little effect on pitch perception. a** Cochlear filter bandwidths were scaled to be two times narrower or two times broader than those estimated for normal-hearing humans. This manipulation is evident in the width of auditory nerve tuning curves measured from five individual fibers per condition (upper left panel). Tuning curves plot thresholds for each fiber as a function of pure tone frequency. Right and lower left panels show simulated auditory nerve representations of the same stimulus (harmonic tone with 200 Hz F0) for each bandwidth condition. Each peripheral representation is flanked by the stimulus power spectrum and the time-averaged auditory nerve excitation pattern. The excitation patterns are altered by changes in frequency selectivity, with coarser tuning yielding less pronounced peaks for individual harmonics, as expected. **b** Cochlear filters modeled on the human ear were replaced with a set of linearly spaced filters with constant bandwidths in Hz. Pure tone tuning curves measured with linearly spaced filters are much sharper than those estimated for humans at higher frequencies (left panel; note the log-spaced frequency scale). The right panel shows the simulated auditory nerve representation of the stimulus from **a** with linearly spaced cochlear filters. In this condition, all harmonics are equally resolved by the cochlear filters and thus equally likely to produce peaks in the time-averaged excitation pattern. **c** Schematic of stimuli used to measure F0 discrimination thresholds. Gray level denotes amplitude. Two example trials are shown, with two different lowest harmonic numbers. **d** F0 discrimination thresholds as a function of lowest harmonic number, measured from networks trained and tested with each of the four peripheral model configurations depicted in **a** and **b**. The best thresholds and the transition points from good to poor thresholds (defined as the lowest harmonic number for which thresholds first exceeded 1%) are re-plotted to the left of and below the main axes, respectively. Lines plot means across the ten networks; error bars indicate 95% confidence intervals bootstrapped across the ten networks.

F0 discrimination behavior (Fig. 6c, d). Broadening filters by a factor of two had no significant effect on the best thresholds ($t(18) = 0.40$, $p = 0.69$, $t$-test comparing thresholds when lowest harmonic number = 1 to the human tuning condition).

Narrowing filters by a factor of two yielded an improvement in best thresholds that was statistically significant ($t(18) = 2.74$, $p = 0.01$, $d = 1.23$) but very small (0.27% vs. 0.32% for the networks with normal human tuning). Linearly spaced cochlear

filters also yielded best thresholds that were not significantly different from those for normal human tuning ($t(18) = 1.88$, $p = 0.08$). In addition, the dependence of thresholds on harmonic number was fairly similar in all cases (Fig. 6d). The transition between good and poor thresholds occurred around the sixth harmonic irrespective of the cochlear bandwidths (not significantly different for any of the three altered tuning conditions: two times broader, $t(18) = 1.33$, $p = 0.20$; two times narrower, $t(18) = 1.00$, $p = 0.33$; linearly spaced, $t(18) = 0.37$, $p = 0.71$; $t$-tests comparing to the normal human tuning condition).

All three models with altered cochlear filter bandwidths produced worse thresholds for stimuli containing only high-numbered harmonics (Fig. 6d). This effect is expected for the narrower and linearly spaced conditions (smaller bandwidths result in reduced envelope cues from beating of adjacent harmonics), but we do not have an explanation for why networks with broader filters also produced poorer thresholds. One possibility that we ruled out is overfitting of the network architectures to the human cochlear filter bandwidths; validation set accuracies were no worse with broader filters ($t(18) = 0.66$, $p = 0.52$). However, we note that all of the models exhibit what would be considered poor performance for stimuli containing only high harmonics (thresholds are at least an order of magnitude worse than they are for low harmonics), and are thus all generally consistent with human perception in this regime.

We also simulated the full suite of psychophysical experiments from Fig. 2 on networks with altered frequency tuning. Most experimental results were robust to peripheral frequency tuning (Supplementary Fig. 7).

**Dependence of pitch behavior on training set sound statistics.** In contrast to the widely debated roles of peripheral cues, the role of natural sound statistics in pitch has been little discussed throughout the history of hearing research. To investigate how optimization for natural sounds may have shaped pitch perception, we fixed the cochlear representation to its normal human settings and instead manipulated the characteristics of the sounds on which networks were trained.

**Altered training set spectra produce altered behavior.** One salient property of speech and instrument sounds is that they typically have more energy at low frequencies than high frequencies (Fig. 7a, left column, black line). To test if this lowpass characteristic shapes pitch behavior, we trained networks on highpass-filtered versions of the same stimuli (Fig. 7a, left column, orange line) and then measured their F0 discrimination thresholds (Fig. 7b). For comparison, we performed the same experiment with lowpass-filtered sounds.

Thresholds measured from networks optimized for highpass sounds exhibited a much weaker dependence on harmonic number than if optimized for natural sounds (Fig. 7c, left column). This difference produced an interaction between the effects of harmonic number and the training condition ($F(2.16, 38.85) = 72.33$, $p < 0.001$, $\eta^2_{partial} = 0.80$). By contrast, the dependence on harmonic number was accentuated for lowpass-filtered stimuli, again producing an interaction between the effects of harmonic number and the training condition ($F(4.25, 76.42) = 30.81$, $p < 0.001$, $\eta^2_{partial} = 0.63$).

We also simulated the full suite of psychophysical experiments on these networks (Supplementary Fig. 8) and observed several other striking differences in their performance characteristics. In particular, networks optimized for highpass-filtered natural sounds exhibited better discrimination thresholds for transposed tones than pure tones ($t(18) = 9.92$, $p < 0.001$, $d = 4.43$, two-sided

two-sample $t$-test comparing pure tone and transposed tone thresholds averaged across frequency), a complete reversal of the human result. These results illustrate that the properties of pitch perception are not strictly a function of the information available in the periphery—performance characteristics can depend strongly on the "environment" in which a system is optimized.

**Natural spectral statistics account for human-like behavior.** To isolate the acoustic properties needed to reproduce human-like pitch behavior, we also trained networks on synthetic tones embedded in masking noise, with spectral statistics matched to those of the natural sound training set (Fig. 7a, center column). Specifically, we fit multivariate Gaussians to the spectral envelopes of the speech/instrument sounds and the noise from the original training set, and synthesized stimuli with spectral envelopes sampled from these distributions. Although discrimination thresholds were overall somewhat better than when trained on natural sounds, the resulting network again exhibited human-like pitch characteristics (Fig. 7c, center column, black line). Because the synthetic tones were constrained only by the mean and covariance of the spectral envelopes of our natural training data, the results suggest that such low-order spectral statistics capture much of the natural sound properties that matter for obtaining human-like pitch perception (see Supplementary Fig. 8 for results on the full suite of psychophysical experiments).

For comparison, we also trained networks on synthetic tones with spectral statistics that deviate considerably from speech and instrument sounds. We generated these "anti-matched" synthetic tones by multiplying the mean of the fitted multivariate Gaussian by negative one (see Methods) and sampling spectral envelopes from the resulting distribution. Training on the resulting highpass synthetic tones (Fig. 7a, center column, orange line) completely reversed the pattern of behavior seen in humans: discrimination thresholds were poor for stimuli containing low-numbered harmonics and good for stimuli containing only high-numbered harmonics (producing a negative correlation with human results: $r = -0.98$, $p < 0.001$, Pearson correlation) (Fig. 7c, center column, orange line). These results further illustrate that the dominance of low-numbered harmonics in human perception is not an inevitable consequence of cochlear transduction—good pitch perception is possible in domains where it is poor in humans, provided the system is trained to extract the relevant information.

**Music-trained networks exhibit better pitch acuity.** We also trained networks separately using only speech or only music stimuli (Fig. 7a, right column). Consistent with the more accurate pitch discrimination found in human listeners with musical training[54], networks optimized specifically for music have lower discrimination thresholds for stimuli with low-numbered harmonics (Fig. 7c, right column; $t(18) = 9.73$, $p < 0.001$, $d = 4.35$, two-sample $t$-test comparing left-most conditions—which produce the best thresholds—for speech and music training). As a test of whether this result could be explained by cochlear processing, we repeated this experiment on networks with learnable first-layer filters (as in Fig. 4) and found that networks optimized specifically for music still produced lower absolute thresholds (Supplementary Fig. 9). This result likely reflects the greater similarity of the synthetic test tones (standardly used to assess pitch perception) to instrument notes compared to speech excerpts, the latter of which are less perfectly periodic over the stimulus duration.

**Training set noise required for "missing fundamental" illusion.** One of the core challenges of hearing is the ubiquity of background

**Fig. 7 Pitch perception depends on training set sound statistics. a** Average power spectrum of training stimuli under different training conditions. Networks were trained on datasets with lowpass- and highpass-filtered versions of the primary speech and music stimuli (column 1), as well as datasets of synthetic tones with spectral statistics either matched or anti-matched (Methods) to those of the primary dataset (column 2), and datasets containing exclusively speech or music (column 3). Filtering indicated in column 1 was applied to the speech and music stimuli prior to their superposition on background noise. Gray shaded regions plot the average power spectrum of the background noise that pitch-evoking sounds were embedded in for training purposes. **b** Schematic of stimuli used to measure F0 discrimination thresholds as a function of lowest harmonic number. Two example trials are shown, with two different lowest harmonic numbers. **c** F0 discrimination thresholds as a function of lowest harmonic number, measured from networks trained on each dataset shown in A. Lines plot means across the ten networks; error bars indicate 95% confidence intervals bootstrapped across the ten networks.

noise. To investigate how pitch behavior may have been shaped by the need to hear in noise, we varied the level of the background noise in our training set. Networks trained in noisy environments (Fig. 8, left) resembled humans in accurately inferring F0 even when the F0 was not physically present in the stimuli (thresholds for stimuli with lowest harmonic number between 2 and 5 were all under 1%). This "missing fundamental illusion" was progressively weakened in networks trained in higher SNRs (Fig. 8, center and right), with discrimination thresholds sharply elevated when the lowest harmonic number exceeded two ($F(2,27) = 6.79$, $p < 0.01$, $\eta^2_{partial} = 0.33$; main effect of training condition when comparing thresholds for lowest harmonic numbers between 2 and 5).

Networks trained in noiseless environments also deviated from human behavior when tested on alternating-phase (Fig. 8b, row 2) and frequency-shifted complexes (Fig. 8b, row 3), apparently ignoring high-numbered harmonics (correlations with human results were lower in both experiments; $t(18) = 9.08$, $p < 0.001$, $d = 4.06$ and $t(18) = 4.41$, $p < 0.001$, $d = 1.97$, comparing high vs. no training noise). Conversely, discrimination thresholds for pure tones (Fig. 8b, row 5) remained good (below 1%), as though the networks learned to focus primarily on the first harmonic. Collectively, these results suggest the ability to extract F0 information from high-numbered harmonics in part reflects an adaptation for hearing in noise.

**Network neurophysiology**. Although our primary focus in this paper was to use DNNs to understand behavior in normative terms, we also examined whether the internal representations of our model might exhibit established neural phenomena.

We simulated electrophysiology experiments on our best-performing network architecture by measuring time-averaged model unit activations to pure and complex tones varying in harmonic composition (Fig. 9a). F0 tuning curves of units in different network layers (Fig. 9b) illustrate a transition from frequency-tuned units in the first layer (relu_0, where units responded whenever a harmonic of a complex tone aligned with their pure tone tuning) to complex tuning in intermediate layers (relu_2, relu_4, and fc_int) to unambiguous F0 tuning in the final layer (fc_top), where units responded selectively to specific F0s across different harmonic compositions. These latter units thus resemble pitch-selective neurons identified in primate auditory cortex[28] in which tuning to the F0 of missing-fundamental complexes aligns with pure tone tuning.

We quantified the F0 tuning of individual units by measuring the correlation between pure tone and complex tone tuning curves. High correlations between tuning curves indicate F0 tuning invariant to harmonic composition. In each of the ten best-performing networks, units became progressively more F0-tuned deeper into the network (Fig. 9c, right, solid symbols). Critically, this result depended on the harmonicity of the tones. When we

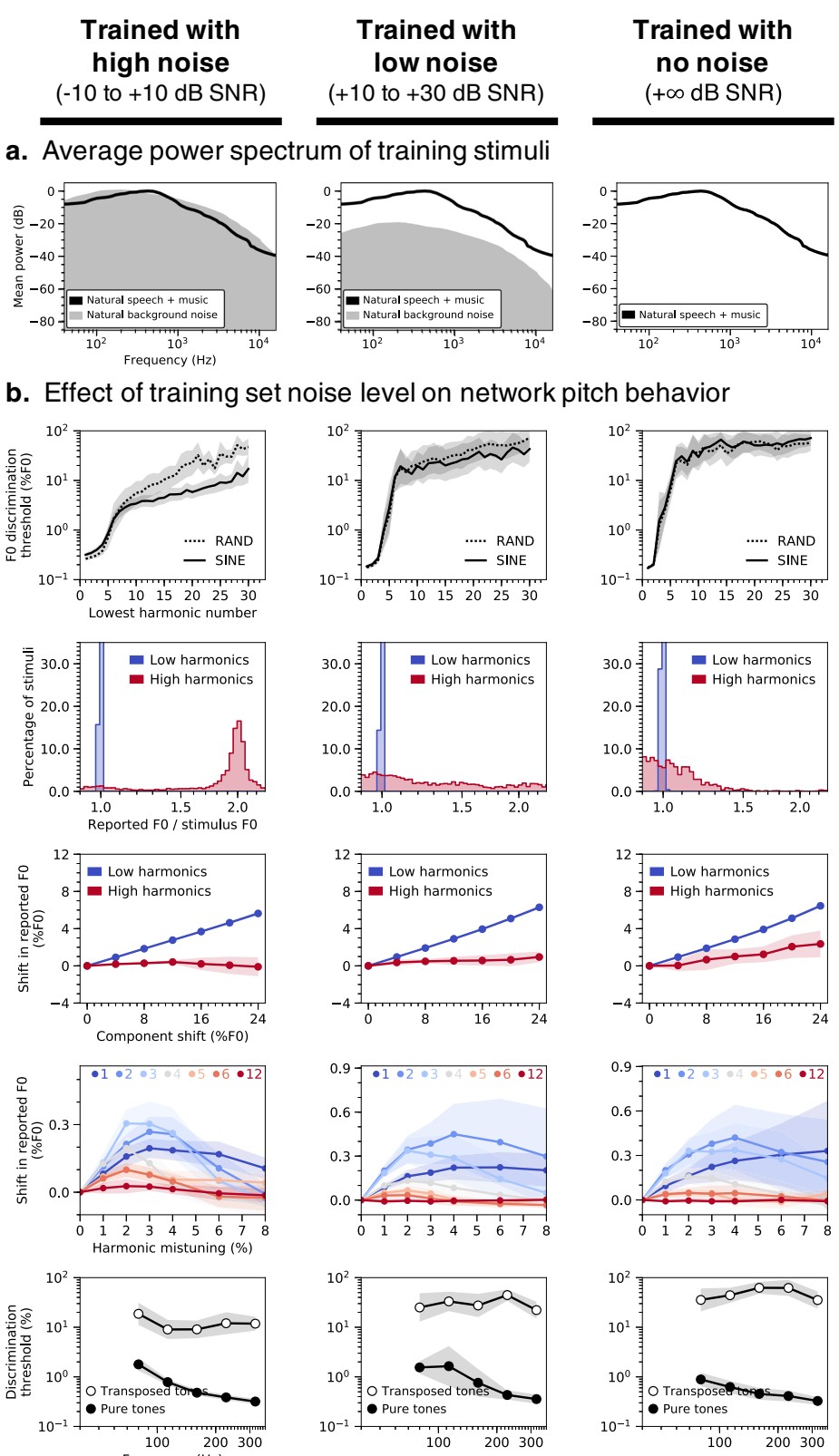

**Fig. 8 Key characteristics of human pitch behavior only emerge in noisy training conditions. a** Average power spectrum of training stimuli. Networks were trained on speech and music stimuli embedded in three different levels of background noise: high (column 1), low (column 2), and none (column 3). **b** Effect of training set noise level on network behavior in all five main psychophysical experiments (see Fig. 2a–e): F0 discrimination as a function of harmonic number and phase (row 1), pitch estimation of alternating-phase stimuli (row 2), pitch estimation of frequency-shifted complexes (row 3), pitch estimation of complexes with individually mistuned harmonics (row 4), and frequency discrimination with pure and transposed tones (row 5). Lines plot means across the ten networks; error bars indicate 95% confidence intervals bootstrapped across the ten networks.

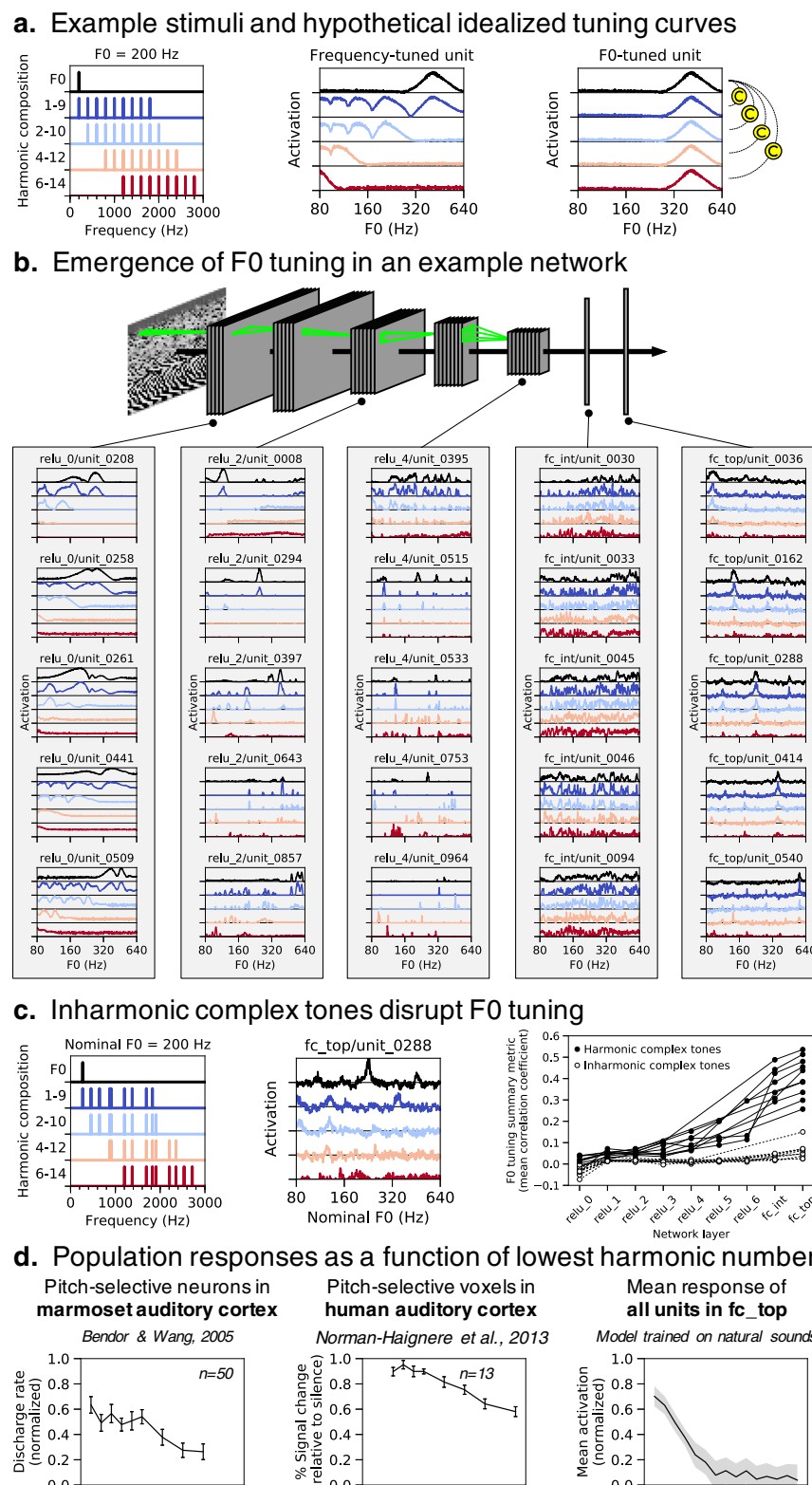

**a.** Example stimuli and hypothetical idealized tuning curves

**b.** Emergence of F0 tuning in an example network

**c.** Inharmonic complex tones disrupt F0 tuning

**d.** Population responses as a function of lowest harmonic number

repeated the analysis with complex tones made inharmonic by jittering component frequencies[20] (Fig. 9c, left), network units no longer showed F0 tuning (Fig. 9b, center) and the dependence on network layer was eliminated (Fig. 9c, right, open symbols). In this respect the units exhibit a signature of human F0-based pitch, which is also disrupted by inharmonicity[20,55], and of pitch-tuned neurons in nonhuman primates[56].

To compare the population tuning to that observed in the auditory system, we also measured unit activations to harmonic complexes as a function of the lowest harmonic in the stimulus. The F0-tuned units in our model's final layer responded more strongly when stimuli contained low-numbered harmonics (Fig. 9d, right; main effect of lowest harmonic number on mean activation, $F(1.99,17.91) = 134.69$, $p < 0.001$, $\eta^2_{partial} = 0.94$). This

**Fig. 9 Network neurophysiology. Network activations were measured in response to pure tones and complex tones with four different harmonic compositions. a** Left: Power spectra for stimuli with 200 Hz F0. Center: expected F0 tuning curves for an idealized frequency-tuned unit. The tuning curves are color-matched to the corresponding stimulus (e.g., black for pure tones and red for harmonics 6–14). A frequency-tuned unit should respond to pure tones near its preferred frequency (414 Hz) or to complex tones containing harmonics near its preferred frequency (e.g., when F0 = 212, 138, 103.5, or 82.8 Hz, i.e., 414/2, 414/3, or 414/4 Hz). Right: expected F0 tuning curves for an idealized F0-tuned unit. An F0-tuned unit should produce tuning curves that are robust to harmonic composition. The strength of a unit's F0 tuning can thus be quantified as the mean correlation between the pure tone (frequency) tuning curve and each of the complex tone tuning curves. **b** F0 tuning curves measured from five representative units in each of five network layers. Units in the first layer (relu_0) seem to exhibit frequency tuning. Units in the last layer (fc_top) exhibit F0 tuning. **c** Left: Nominal F0 tuning curves were measured for complex tones made inharmonic by jittering component frequencies. Center: Such curves are shown for one example unit in the network's last layer. Unlike for harmonic tones, the tuning curves for tones with different frequency compositions do not align. Right: The overall F0 tuning of a network layer was computed by averaging the F0 tuning strength across all units in the layer. A unit's F0 tuning strength was quantified as the mean correlation between the pure tone (frequency) tuning curve and each of the complex tone tuning curves. For each of our ten best network architectures, overall F0 tuning (computed separately using either harmonic or inharmonic complex tones) is plotted as a function of network layer. Network units become progressively more F0-tuned deeper into the networks, but only for harmonic tones. **d** Left: Population responses of pitch-selective units in marmoset auditory cortex, human auditory cortex, and our model's output layer, plotted as a function of lowest harmonic number. Marmoset single-unit recordings were made from three animals and error bars indicate SEM across 50 neurons (re-plotted from[28]). Center: Human fMRI responses to harmonic tones, as a function of their lowest harmonic number. Data were collected from 13 participants and error bars indicate within-subject SEM (re-plotted from[29]). Responses were measured from a functional region of interest defined by a contrast between harmonic tones and frequency-matched noise. Responses were measured in independent data (to avoid double dipping). Right: Network unit activations to harmonic tones as a function of lowest harmonic number. Activations were averaged across all units in the final fully connected layer of our ten best network architectures (error bars indicate 95% confidence intervals bootstrapped across the ten best network architectures).

result mirrors the response characteristics of pitch-selective neurons (measured with single-unit electrophysiology) in marmoset auditory cortex (Fig. 9d, left)[28] and pitch-selective voxels (measured with fMRI) in human auditory cortex (Fig. 9d, center)[29].

## Discussion

We developed a model of pitch perception by optimizing artificial neural networks to estimate the fundamental frequency of their acoustic input. The networks were trained on simulated auditory nerve representations of speech and music embedded in background noise. The best-performing networks closely replicated human pitch judgments in simulated psychophysical experiments despite never being trained on the psychophysical stimuli. To investigate which aspects of the auditory periphery and acoustical environment contribute to human-like pitch behavior, we optimized networks with altered cochleae and sound statistics. Lowering the upper limit of phase locking in the auditory nerve yielded models with behavior unlike that of humans: F0 discrimination was substantially worse than in humans and had a distinct dependence on stimulus characteristics. Model behavior was substantially less sensitive to changes in cochlear frequency tuning. However, the results were also strongly dependent on the sound statistics the model was optimized for. Optimizing for stimuli with unnatural spectra, or without concurrent background noise yielded behavior qualitatively different from that of humans. The results suggest that the characteristics of human pitch perception reflect the demands of estimating the fundamental frequency of natural sounds, in natural conditions, given a human cochlea.

Our model innovates on prior work in pitch perception in two main respects. First, the model was optimized to achieve accurate pitch estimation in realistic conditions. By contrast, most previous pitch models have instantiated particular mechanistic or algorithmic hypotheses[25,33–40]. Our model's initial stages incorporated detailed simulations of the auditory nerve, but the rest of the model was free to implement any of a wide set of strategies that optimized performance. Optimization enabled us to test normative explanations of pitch perception that have previously been neglected. Second, the model achieved reasonable quantitative matches to human pitch behavior. This match to behavior allowed strong tests of the role of different elements of peripheral

coding in the auditory nerve. Prior work attempted to derive optimal decoders of frequency from the auditory nerve[9,49], but was unable to assess pitch perception (i.e., F0 estimation) due to the added complexity of this task.

Both of these innovations were enabled by contemporary "deep" neural networks. For our purposes, DNNs instantiate general-purpose functions that can be optimized to perform a training task. They learn to use task-relevant information present in the sensory input, and avoid the need for hand-designed methods to extract such information. This generality is important for achieving good performance on real-world tasks. Hand-designed models, or simpler model classes, would likely not provide human-level performance. For instance, we found that very shallow networks both produced worse overall performance, and a poorer match to human behavior (Supplementary Fig. 3).

Although mechanistic explanations of pitch perception are widely discussed[33–38,40], there have been few attempts to explain pitch in normative terms. But like other aspects of perception, pitch is plausibly the outcome of an optimization process (realized through some combination of evolution and development) that produces good performance under natural conditions. We found evidence that these natural conditions have a large influence on the nature of pitch perception, in that human-like behavior emerged only in models optimized for naturalistic sounds heard in naturalistic conditions (with background noise).

In particular, the demands of extracting the F0 of natural sounds appear to explain one of the signature characteristics of human pitch perception: the dependence on low-numbered harmonics. This characteristic has traditionally been proposed to reflect limitations of cochlear filtering, with filter bandwidths determining the frequencies that can be resolved in a harmonic sound[22,43,48,52]. However, we found that the dependence on harmonic number could be fully reversed for sufficiently unnatural sound training sets (Fig. 7c). Moreover, the dependence was stable across changes in cochlear filter bandwidths (Fig. 6c). These results suggest that pitch characteristics primarily reflect the constraints of natural sound statistics (specifically, lowpass power spectra) coupled with the high temporal fidelity of the auditory nerve. In the language of machine learning, discrimination thresholds appear to partly be a function of the match between the test stimuli and the training set (i.e., the sensory signals a perceptual system was optimized for). Our

results suggest that this match is critical to explaining many of the well-known features of pitch perception.

A second influence of the natural environment was evident when we eliminated background noise from the training set (Fig. 8). Networks trained without background noise did not extract F0 information from high-numbered harmonics, relying entirely on the lowest-numbered harmonics. Such a strategy evidently works well for idealized environments (where the lowest harmonics are never masked by noise), but not for realistic environments containing noise, and diverges from the strategy employed by human listeners. This result suggests that pitch is also in part a consequence of needing to hear in noise, and is consistent with evidence that human pitch perception is highly noise-robust[57]. Together, these two results suggest that explanations of pitch perception cannot be separated from the natural environment.

The approach we propose here contrasts with prior work that derived optimal strategies for psychophysical tasks on synthetic stimuli[9,49,58,59]. Although human listeners often improve on such tasks with practice, there is not much reason to expect humans to approach optimal behavior for arbitrary tasks and stimuli (because these do not drive natural selection, or learning during development). By contrast, it is plausible that humans are near-optimal for important tasks in the natural environment, and that the consequences of this optimization will be evident in patterns of psychophysical performance, as we found here.

Debates over pitch mechanisms have historically been couched in terms of the two axes of the cochlear representation: place and time. Place models analyze the signature of harmonic frequency spectra in the excitation pattern along the length of the cochlea[34,35], whereas temporal models quantify signatures of periodicity in temporal patterns of spikes[33,37]. Our model makes no distinction between place and time per se, using whatever information in the cochlear representation is useful for the training task. However, we were able to assess its dependence on peripheral resolution in place and time by altering the simulated cochlea. These manipulations provided evidence that fine-grained peripheral timing is critical for normal pitch perception (Fig. 5c, e), and that fine-grained place-based frequency tuning is less so (Fig. 6). Some degree of cochlear frequency selectivity is likely critical to enabling phase locking to low-numbered harmonics, but such effects evidently do not depend sensitively on tuning bandwidth. These conclusions were enabled by combining a realistic model of the auditory periphery with task-optimized neural networks.

Our model is consistent with most available pitch perception data, but it is not perfect. For instance, the inflection point in the graph of Fig. 2a occurs at a somewhat lower harmonic number in the model than in humans. Given the evidence presented here that pitch perception reflects the stimulus statistics a system is optimized for, some discrepancies might be expected from the training set, which (due to the limitations of available corpora) consisted entirely of speech and musical instrument sounds, and omitted other types of natural sounds that are periodic in time. The range of F0s we trained on was similarly limited by available audio datasets, and prevents us from making predictions about the perception of very high frequencies[60]. The uniform distributions over sound level and SNR in our training dataset were also not matched in a principled way to the natural world. Discrepancies may also reflect shortcomings of our F0 estimation task (which used only 50 ms clips) or peripheral model, which although state-of-the-art and relatively well validated, is imperfect (e.g., peripheral representations consisted of firing rates rather than spikes).

We note that the ear itself is the product of evolution and thus likely itself reflects properties of the natural environment[61]. We chose to train models on a fixed representation of the ear in part to address longstanding debates over the role of established features of peripheral neural coding on pitch perception. We view this approach as sensible on the grounds that the evolution of the cochlea was plausibly influenced by many different natural behaviors, such that it is more appropriately treated as a constraint on a model of pitch rather than a model stage to be derived along with the rest of the model. Consistent with this view, when we replaced the fixed peripheral model with a set of learnable filters operating directly on sound waveforms, networks exhibited less human-like pitch behavior (Fig. 4). This result suggests it could be fruitful to incorporate additional stages of peripheral physiology, which might similarly provide constraints on pitch perception.

Our model shares many of the commonly-noted limitations of DNNs as models of the brain[62,63]. Our optimization procedure is not a model of biological learning and/or evolution, but rather provides a way to obtain a system that is optimized for the training conditions given a particular peripheral representation of sound. Biological organisms are almost certainly not learning to estimate F0 from thousands of explicitly labeled examples, and in the case of pitch may leverage their vocal ability to produce harmonic stimuli to hone their perceptual mechanisms. These differences could cause the behavior of biological systems to deviate from optimized neural networks in some ways.

The neural network architectures we used here are also far from fully consistent with biology, being only a coarse approximation to neural networks in the brain. Although similarities have been documented between trained neural network representations and brain representations[16,18], and although we saw some such similarities ourselves in the network's activations (Fig. 9c), the inconsistencies with biology could lead to behavioral differences compared to humans.

And although our approach is inspired by classical ideal observer models, the model class and optimization methods likely bias the solutions to some extent, and are not provably optimal like classic ideal observer models. Nonetheless, the relatively good match to available data suggests that the optimization is sufficiently successful as to be useful for our purposes.

The model developed here performs a single task—that of estimating the F0 of a short sound. Human pitch behavior is often substantially more complex, in part because information is conveyed by how the F0 changes over time, as in prosody[64] or melody[65]. In some cases relative pitch involves comparisons of the spectrum rather than the F0[20,55] and/or can be biased by changes in the timbre of a sound[66], for reasons that are not well understood. The framework used here could help to develop normative understanding of such effects, by incorporating more complicated tasks (e.g., involving speech or music) and then characterizing the pitch-related behavior that results. DNNs that perform more complex pitch tasks might also exhibit multiple stages of pitch representations that could provide insight into putative hierarchical stages of auditory cortex[18,27,67].

The approach we used here has natural extensions to understanding other aspects of hearing[68], in which similar questions about the roles of peripheral cues have remained unresolved. Our methods could also be extended to investigate hearing impairment, which can be simulated with alterations to standard models of the cochlea[69] and which often entails particular types of deficits in pitch perception[51]. Prostheses such as cochlear implants are another natural application of task-optimized modeling. Current implants restore some aspects of hearing relatively well, but pitch perception is not one of them[70]. Models optimized with different types of simulated electrical stimulation could clarify the patterns of behavior to expect. Models trained with either acoustically- or electrically-stimulated peripheral auditory representations (or

combinations thereof) and then tested with electrically-stimulated input could yield insights into the variable outcomes of pediatric cochlear implantation. Similar approaches could be applied to study acclimatization to hearing aids in adults.

There is also growing evidence for species differences in pitch perception[71,72]. Our approach could be used to relate species differences in perception to species differences in the cochlea[73] or to differences in the acoustic environment and/or tasks a species may be optimized for. While our results suggest that differences in cochlear filters alone are unlikely to explain differences in pitch perception abilities across species, they leave open the possibility that human pitch abilities reflect the demands of speech and music, which plausibly require humans to be more sensitive to small F0 differences than other species. This issue could be clarified by optimizing network representations for different auditory tasks.

More generally, the results here illustrate how supervised machine learning enables normative analysis in domains where traditional ideal observers are intractable, an approach that is broadly applicable outside of pitch and audition.

## Methods

**Natural sounds training dataset—overview.** The main training set consisted of 50 ms excerpts of speech and musical instruments. This duration was chosen to enable accurate pitch perception in human listeners[74], but to be short enough that the F0 would be relatively stable even in natural sounds such as speech that have time-varying F0s. The F0 label for a training example was estimated from a "clean" speech or music excerpt. These excerpts were then superimposed on natural background noise. Overall stimulus presentation levels were drawn uniformly between 30 dB SPL and 90 dB SPL. All training stimuli were sampled at 32 kHz.

**Speech and music training excerpts.** We used STRAIGHT[75] to compute time-varying F0 and periodicity traces for sounds in several large corpora of recorded speech and instrumental music: Spoken Wikipedia Corpora (SWC)[76], Wall Street Journal (WSJ), CMU Kids Corpus, CSLU Kids Speech, NSynth[77], and RWC Music Database. STRAIGHT provides accurate estimates of the F0 provided the background noise is low, as it was in each of the corpora. Musical instrument recordings were notes from the chromatic scale, and thus were spaced roughly in semitones. To ensure that sounds would span a continuous range of F0s, we randomly pitch-shifted each instrumental music recording by a small amount (up to ±3% F0, via resampling).

Source libraries were constructed for each corpus by extracting all highly periodic (time-averaged periodicity level > 0.8) and non-overlapping 50ms segments from each recording. We then generated our natural sounds training dataset by sampling segments with replacement from these source libraries to uniformly populate 700 log-spaced F0 bins between 80 Hz and 1000 Hz (bin width = 1/16 semitones = 0.36% F0). Segments were assigned to bins according to their time-averaged F0. The resulting training dataset consisted of 3000 exemplars per F0 bin for a total of 2.1 million exemplars. The relative contribution of each corpus to the final dataset was constrained both by the number of segments per F0 bin available in each source library (the higher the F0, the harder it is to find speech clips) and the goal of using audio from many different speakers, instruments, and corpora. The composition we settled on is:

- F0 bins between 80 Hz and 320 Hz

  50% instrumental music (1000 NSynth and 500 RWC clips per bin)
  50% adult speech (1000 SWC and 500 WSJ clips per bin)

- F0 bins between 320 Hz and 450 Hz

  50% instrumental music (1000 NSynth and 500 RWC clips per bin)
  50% child speech (750 CSLU and 750 CMU clips per bin)

- F0 bins between 450 Hz and 1000 Hz

  100% instrumental music (2500 NSynth and 500 RWC clips per bin)

**Background noise for training data.** To make the F0 estimation task more difficult and to simulate naturalistic listening conditions, each speech or instrument excerpt in the training dataset was embedded in natural background noise. The signal-to-noise ratio for each training example was drawn uniformly between −10 dB and +10 dB. Noise source clips were taken from a subset of the AudioSet corpus[78], screened to remove nonstationary sounds (e.g., speech or music). The

screening procedure involved measuring auditory texture statistics (envelope means, correlations, and modulation power in and across cochlear frequency channels)[79] from all recordings, and discarding segments over which these statistics were not stable in time, as in previous studies[80]. To ensure the F0 estimation task remained well defined for the noisy stimuli, background noise clips were also screened for periodicity by computing their autocorrelation functions. Noise clips with peaks greater than 0.8 at lags greater than 1 ms in their normalized auto-correlation function were excluded.

**Peripheral auditory model.** The Bruce et al. (2018) auditory nerve model was used to simulate the peripheral auditory representation of every stimulus. This model was chosen because it captures many of the complex response properties of auditory nerve fibers and has been extensively validated against electro-physiological data from cats[42,69]. Stages of peripheral signal processing in the model include: a fixed middle-ear filter, a nonlinear cochlear filter bank to simulate level-dependent frequency tuning of the basilar membrane, inner and outer hair cell transduction functions, and a synaptic vesicle release/re-docking model of the synapse between inner hair cells and auditory nerve fibers. Although the model's responses have only been directly compared to recordings made in nonhuman animals, some model parameters have been inferred for humans (such as the bandwidths of cochlear filters) on the basis of behavioral and otoacoustic measurements[53].

Because the majority of auditory nerve fibers, especially those linked to feedforward projections to higher auditory centers, have high spontaneous firing rates[81,82], we used exclusively high spontaneous rate fibers (70 spikes/s) as the input to our model. To control for the possibility that spontaneous auditory nerve fiber activity could influence pitch behavior (for instance, at conversational speech levels, firing rates of high spontaneous rate fibers are typically saturated, which may degrade excitation pattern cues to F0), we additionally trained and tested the ten best-performing networks from the architecture search using exclusively low spontaneous rate fibers (0.1 spikes/s). The average results for these networks are shown in Supplementary Fig. 10. We found that psychophysical behavior was qualitatively unaffected by nerve fiber spontaneous rate. These results suggested to us that high spontaneous rate fibers were sufficient to yield human-like pitch behavior, so we exclusively used high spontaneous rate fibers in all other experiments.

In most cases, the input to the neural network models consisted of the instantaneous firing rate responses of 100 auditory nerve fibers with characteristic frequencies spaced uniformly on an ERB-number scale[83] between 125 Hz and 14,000 Hz. Firing rates were used to approximate the information that would be available in a moderate group of spiking nerve fibers receiving input from the same inner hair cell. The use of 100 frequency channels primarily reflects computational constraints (CPU time for simulating peripheral representations, storage costs, and GPU memory for training), but we note that this number is similar to that used in other auditory models with cochlear front-ends[84]. We confirmed that increasing the number of channels by a factor of ten had little effect on the behavioral results from our main natural sound training condition (Supplementary Figs. 4 and 5), and given that 100 channels were sufficient to obtain model thresholds on par with those of humans, it appears that there is little benefit to additional channels for the task we studied.

To prevent the stimuli being dominated by sound onset/offset effects, each stimulus was padded with 100 ms of the original waveform before being passed through the nerve model. The resulting 150 ms auditory nerve responses were resampled to 20 kHz. The middle 50 ms was then excerpted, leaving a 100-fiber by 1000-timestep array of instantaneous firing rates that constituted the input to the neural networks.

**Deep neural network models—overview.** The 100-by-1000 simulated auditory nerve representations were passed into deep convolutional neural networks, each consisting of a series of feedforward layers. These layers were hierarchically organized and instantiated one of a number of simple operations: linear convolution, pointwise nonlinear rectification, weighted average pooling, batch normalization, linear transformation, dropout regularization, and softmax classification.

The last layer of each network performed F0 classification. We opted to use classification with narrow F0 bins rather than regression in order to soften the assumption that output F0 distributions for a stimulus should be unimodal. For example, an octave error would incur a very large penalty under standard regression loss functions (e.g., L1 or L2), which measure the distance between the predicted and target F0. Classification loss functions, such as the softmax cross-entropy used here, penalize all misclassifications equally. In preliminary work, we found classification networks were empirically easier to train than regression networks and yielded smaller median F0 errors.

The precision of the network's F0 estimate is limited by the bin width of the output layer (and by the precision of the training set labels). We chose a bin width of 1/16 semitones (0.36%). We found empirically that the median F0 estimation error increased for bins wider than this value, and did not improve for narrower bins (Supplementary Fig. 11A). This performance asymptote could reflect the limits of the F0 labels the network was trained on. As it happened, with this bin width of 1/16 of a semitone it was possible to attain discrimination thresholds for synthetic

tones that were on par with the best thresholds typically measured in human listeners (~0.1–0.4%)[43,48] for some model architectures and auditory nerve settings. Discrimination thresholds were worse for wider classification bins (Supplementary Fig. 11B). We otherwise observed no qualitative change in the pattern of psychophysical results as the bin width was changed. The bin width might thus be considered analogous to decision noise that is sometimes added to models to match human performance (though our choice of bin width appears near-optimal for the dataset we worked with). We note that discrimination thresholds for synthetic tones were also plausibly limited by the similarity of the tones to the training data.

### Definitions of constituent neural network operations

*Convolutional layer.* A convolutional layer implements the convolution of a bank of $N_k$ two-dimensional linear filter kernels with an input $X$. Convolution performs the same operation at each point in the input, which for the 2D auditory nerve representations we used entails convolving in both time and frequency. Convolution in time is a natural choice for models of sensory systems as their input has translation-invariant temporal statistics. Because translation invariance does not hold for the frequency dimension, convolution in frequency is less obviously a natural model constraint. However, many types of sound signals are well described by approximate translation invariance in local neighborhoods of the frequency domain, and classical auditory models can often be described as imposing convolution in frequency[84,85]. Moreover, imposing convolution greatly reduces the number of parameters to be learned. We have empirically found that auditory neural network models often train more readily when convolution in frequency is imposed, suggesting that it is a useful form of model regularization.

The input is a three-dimensional array with shape $[M_f, M_t, M_k]$. For the first convolutional layer in our networks, the input shape was $[100, 1000, 1]$, corresponding to 100 frequency bins (nerve fibers), 1000 timesteps, and a placeholder 1 in the filter kernel dimension.

A convolutional layer is defined by five parameters:

1. $h$: height of the convolutional filter kernels (number of filter taps in the frequency dimension)
2. $w$: width of the convolutional filter kernels (number of filter taps in the time dimension)
3. $N_k$: number of different convolutional filter kernels
4. $W$: Trainable weights for each of the $N_k$ filter kernels; $W$ has shape $[h, w, M_k, N_k]$
5. $B$: Trainable bias vector with shape $[N_k]$

The output of the convolutional layer $Y$ has shape $[N_f, N_t, N_k]$ and is given by:

$$Y[n_f, n_t, n_k] = B[n_k] + \sum_{i=1, j=1, m_k=1}^{h,w,M_k} W[i, j, m_k n_k] \odot X[n_f + i - \lfloor h/2 \rfloor, n_t + j - \lfloor w/2 \rfloor, m_k]$$

where $\odot$ denotes pointwise multiplication and $\lfloor \cdot \rfloor$ denotes integer division. Convolutional layers all used a stride of 1 (i.e., non-strided convolution) and "valid" padding, meaning filters were only applied at positions where every element of the kernel overlapped the input. Due to this boundary handling, the frequency and time dimensions of the output were smaller than those of the input: $N_f = M_f - h + 1$ and $N_t = M_t - w + 1$.

*Pointwise nonlinear rectification.* To learn a nonlinear function, a neural network must contain nonlinear operations. We incorporate nonlinearity via the rectified linear unit (ReLU) activation function, which is applied pointwise to every element $x$ in some input $X$:

$$ReLU(x) = \begin{cases} x & x > 0 \\ 0 & x \leq 0 \end{cases}$$

*Weighted average pooling.* Pooling operations reduce the dimensionality of inputs by aggregating information across adjacent frequency and time bins. To reduce aliasing in our networks (which would otherwise occur from downsampling without first lowpass-filtering), we used weighted average pooling with Hanning windows[62]. This pooling operation was implemented as the strided convolution of a two-dimensional (frequency-by-time) Hanning filter kernel $H$ with an input $X$:

$$Y = H *_{s_f, s_t} X$$

where $*$ denotes convolution and $s_f$ and $s_t$ indicate the stride length in frequency and time, respectively. The Hanning window $H$ had a stride-dependent shape $[h_f, h_t]$, where

$$h_f = \begin{cases} 1 & s_f = 1 \\ 4 \cdot s_f & s_f > 1 \end{cases} \quad and \quad h_t = \begin{cases} 1 & s_t = 1 \\ 4 \cdot s_t & s_t > 1 \end{cases}$$

For an input $X$ with shape $[N_f, N_t, N_k]$, the shape of the output $Y$ is $[N_f/s_f, N_t/s_t, N_k]$. Note that when either $s_f$ or $s_t$ is set to 1, there is no pooling along the corresponding dimension.

*Batch normalization.* Batch normalization is an operation that normalizes its inputs in a pointwise manner using running statistics computed from every batch of training data. Normalizing activations between layers greatly improves DNN

training efficiency by reducing the risk of exploding and vanishing gradients: small changes to network parameters in one layer are less likely to be amplified in successive layers if they are separately normalized. For every batch of inputs $B$ during training, the pointwise batch mean ($\mu_B$) and batch variance ($\sigma_B^2$) are computed and then used to normalize each input $X_b \in B$:

$$X_{b,normalized} = \gamma \left( \frac{X_b - \mu_B}{\sqrt{\sigma_B^2 + \epsilon}} \right) + \beta$$

where all operations are applied pointwise, $\epsilon = 0.001$ to prevent division by zero, and $\gamma$ and $\beta$ are learnable scale and offset parameters. Throughout training, single-batch statistics are used to update the running mean ($\mu_{total}$) and variance ($\sigma_{total}^2$). During evaluation mode, $X_{b,normalized}$ is computed using $\mu_{total}$ and $\sigma_{total}^2$ in place of $\mu_B$ and $\sigma_B^2$.

*Fully connected layer.* A fully connected (or dense) layer applies a linear transformation to its input without any notion of localized frequency or time. An input $X$ with shape $[N_f, N_t, N_k]$, is first reshaped to a vector $X_{flat}$ with shape $[N_f \cdot N_t \cdot N_k]$. Then, $X_{flat}$ is linearly transformed to give an output $Y$ with shape $[N_{out}]$:

$$Y_{out}[n_{out}] = B[n_{out}] + \sum_{n_{in}=1}^{N_f \cdot N_t \cdot N_{ch}} W[n_{out}, n_{in}] \cdot X_{flat}[n_{in}]$$

where $B$ is a bias vector with shape $[N_{out}]$ and $W$ is a weight matrix with shape $[N_{out}, N_{in}]$. The values of $B$ and $W$ are learned during the optimization procedure.

*Dropout regularization.* The dropout operation receives as input a vector $X$ with shape $[N_{in}]$ and randomly selects a fraction ($r$) of its values to set to zero. The remaining values are scaled by $1/(1 - r)$, so that the expected sum over all outputs is equal to the expected sum over all inputs. The $r \cdot N_{in}$ positions in $X$ that get set to zero are chosen at random for every new batch of data. Dropout is commonly used to reduce overfitting in artificial neural networks. It can be thought of as a form of model averaging across the exponentially many sub-networks generated by zeroing-out different combinations of units. All of our networks contained exactly one dropout operation immediately preceding the final fully connected layer. We used a dropout rate of 50% during both training and evaluation.

*Softmax classifier.* The final operation of every network is a softmax activation function, which receives as input a vector $X$ of length $N_{classes}$ (equal to the number of output classes; 700 in our case). The input vector is passed through a normalized exponential function to produce a vector $Y$ of the same length:

$$Y[n_{out}] = \frac{\exp(X[n_{out}])}{\sum_{n_{classes}=1}^{N_{classes}} \exp(X[n_{classes}])}$$

The values of the output vector are all greater than zero and sum to one. $Y$ can be interpreted as a probability distribution over F0 classes for the given input sound.

### Model optimization—architecture search

All of our DNN architectures had the general form of one to eight convolutional layers plus one to two fully connected layers. Each convolutional layer was always immediately followed by three successive operations: ReLU activation function, weighed average pooling, and batch normalization. Fully connected layers were always situated at the end of the network, after the last convolution-ReLU-pooling-normalization block. The final fully connected layer was always immediately followed by the softmax classifier. For architectures with two fully connected layers, the first fully connected layer was followed by a ReLU activation function and a batch normalization operation. In our analyses, we sometimes grouped networks by their number of convolutional layers (e.g., single vs. multi-convolutional-layer networks; Supplementary Fig. 3) regardless of the number of fully connected layers. When we refer to network "activations" in a given convolutional layer (Fig. 9), we always mean the outputs of the ReLU activation function immediately following that convolutional layer.

Within the family of models considered, we generated 400 distinct DNN architectures by randomly sampling from a large space of hyperparameters. The number of convolutional layers was first uniformly drawn from 1 to 8. Within each layer, the number and dimensions of convolutional filter kernels were then sampled based on the size of the layer's input. The number of filter kernels in the first layer was 16, 32, or 64 (each sampled with probability = 1/3). The number of kernels in each successive layer could increase by a factor of 2 (probability = 1/2), stay the same (probability = 1/3), or decrease by a factor of 2 (probability = 1/6) relative to the previous layer. Frequency dimensions of the filter kernels were integers sampled uniformly between 1 and $N_f/2$, where $N_f$ is the frequency dimension of the layer's input. Time dimensions of the filter kernels were integers sampled uniformly between $N_t/20$ and $N_t/2$, where $N_t$ is the time dimension of the layer's input. These sampling ranges tended to produce rectangular filters (longer in the time dimension than the frequency dimension), especially in the early layers. We felt this was a reasonable design choice given the rectangular dimensions of the input (100-by-1000, frequency-by-time). To limit the memory

footprint of the generated DNNs, we imposed 16 and 1024 as lower and upper bounds on the number of kernels in a single layer and capped the frequency × time area of convolutional filter kernels at 256.

The stride lengths for the weighted average pooling operations after each convolutional layer were also sampled from distributions. Pooling stride lengths were drawn uniformly between 1 and 4 for the frequency dimension and 1 and 8 for the time dimension. The existence (probability = 1/2) and size (128, 256, 512, or 1024 units) of a penultimate fully connected layer were also randomly sampled. The final fully connected layer always contained 700 units to support classification into the 700 F0 bins.

**Model optimization—network training**. All 400 network architectures were trained to classify F0 of our natural sounds dataset via stochastic gradient descent with gradients computed via back-propagation. We used a batch size of 64 and the ADAM optimizer with a learning rate of 0.0001. Network weights were trained using 80% of the dataset, and the remaining 20% was held-out as a validation set. Performance on the validation set was measured every 5000 training steps and, to reduce overfitting, training was stopped once classification accuracy stopped increasing by at least 0.5% over 5000 training steps. Training was also stopped for networks that failed to achieve 5% classification accuracy after 10,000 training steps. Each network was able to reach these early-stopping criteria in less than 48 h when trained on a single NVIDIA Tesla V100 GPU.

To ensure conclusions were not based on the idiosyncrasies of any single DNN architecture, we selected the ten architectures that produced the highest validation set accuracies to use as our model experimental "participants" (collectively referred to as "the model"). We re-trained all ten architectures for each manipulation of the peripheral auditory model (Figs. 4–6) and the training set sound statistics (Figs. 7 and 8). The ten different network architectures are described in Supplementary Table 1.

**Network psychophysics—overview**. To investigate network pitch behavior, we simulated a set of classic psychophysical experiments on all trained networks. The general procedure was to (1) pass each experimental stimulus through a network, (2) compute F0 discrimination thresholds or shifts in the "perceived" F0 (depending on the experiment) from network predictions, and (3) compare network results to published data from human listeners tested on the same stimulus manipulations. We selected five psychophysical experiments. These experiments are denoted A through E in the following sections to align with Fig. 2 (which contains schematics of the stimulus manipulations in each experiment). We attempted to reproduce stimuli from these studies as closely as possible, though some modifications were necessary (e.g., all stimuli were truncated to 50 ms to accommodate the input length for the networks). Because the cost of running experiments on networks is negligible, networks were tested on many more (by 1–3 orders of magnitude) stimuli than were human participants. Human data from the original studies was obtained either directly from the original authors (Experiments A, B) or by extracting data points from published figures (Experiments C–E) using Engauge Digitizer (http://markummitchell.github.io/engauge-digitizer). In most cases, individual subject data was not available (the original studies were performed 17–36 years prior to this work), so we report only across-subject means and do not include error bars for human data.

**Experiment A: effect of harmonic number and phase on pitch discrimination**. Experiment A reproduced the stimulus manipulation of Bernstein and Oxenham (2005) to measure F0 discrimination thresholds as a function of lowest harmonic number and phase.

*Stimuli*. Stimuli were harmonic complex tones, bandpass-filtered and embedded in masking noise to control the lowest audible harmonic, and whose harmonics were in sine or random phase. In the original study, the bandpass filter was kept fixed while the F0 was roved to set the lowest harmonic number. Here, to measure thresholds at many combinations of F0 and lowest harmonic number, we roved both the F0 and the location of the filter. We took the 4th-order Butterworth filter (2500–3500 Hz −3 dB passband) described in the original study and translated its frequency response along the frequency axis to set the lowest audible harmonic for a given stimulus. Before filtering, the level of each individual harmonic was set to 48.3 dB SPL, which corresponds to 15 dB above the masked thresholds of the original study's normal-hearing participants. After filtering, harmonic tones were embedded in modified uniform masking noise[48], which has a spectrum that is flat (15 dB/Hz SPL) below 600 Hz and rolls off at 2 dB/octave above 600 Hz. This noise was designed to ensure that only harmonics within the filter's −15 dB passband were audible.

*Human experiment*. The human F0 discrimination thresholds (previously published by Bernstein and Oxenham) were measured from five normal-hearing participants (three female) between the ages of 18 and 21 years old, all self-described amateur musicians with at least 5 years of experience[25]. Each participant completed four adaptive tracks per condition (where a condition had a particular lowest harmonic number and either random or sine phase). Bernstein and Oxenham (2005) reported very similar F0 discrimination thresholds for two

different spectral conditions ("low spectrum" with 2500–3500 Hz filter passband and "high spectrum" with 5000–7000 Hz filter passband). To simplify presentation and because our network experiment measured average thresholds across a wide range of bandpass filter positions, here we report their human data averaged across spectral condition.

*Model experiment*. The F0 discrimination experiment we ran on each network had 600 conditions corresponding to all combinations of two harmonic phases (sine or random), 30 lowest harmonic numbers ($n_{low} = 1, 2, 3 \ldots 30$), and ten reference F0s ($F_{0,ref}$) spaced uniformly on a logarithmic scale between 100 and 300 Hz. Within each condition, each network was evaluated on 121 stimuli with slightly different F0s (within ±6% of $F_{0,ref}$) but the same bandpass filter. The filter was positioned such that the low-frequency cutoff of its −15 dB passband was equal to the frequency of the lowest harmonic for that condition and F0 ($n_{low} \times F_{0,ref}$). On the grounds that human listeners likely employ a strong prior that stimuli should have fairly similar F0s within single trials of a pitch discrimination experiment, we limited network F0 predictions to fall within a one-octave range (centered at $F_{0,ref}$). We simulated a two-alternative forced choice paradigm by making all 7260 possible pairwise comparisons between the 121 stimuli for a condition. In each trial, we asked if the network predicted a higher F0 for the stimulus in the pair with the higher F0 (i.e., if the network correctly identified which of two stimuli had a higher F0). A small random noise term was used to break ties when the network predicted the same F0 for both stimuli. We next constructed a psychometric function by plotting the percentage of correct trials as a function of %F0 difference between two stimuli. We then averaged psychometric functions across the ten reference F0s with the same harmonic phase and lowest harmonic number. Network thresholds were thus based on 1210 stimuli (72,600 pairwise F0 discriminations) per condition. Normal cumulative distribution functions were fit to the 60 resulting psychometric functions (2 phase conditions × 30 lowest harmonic numbers). To match human F0 discrimination thresholds, which were measured with a 2-down-1-up adaptive algorithm, we defined the network F0 discrimination threshold as the F0 difference (in percent, capped at 100%) that yielded 70.7% of trials correct.

*Human-model comparison*. We quantified the similarity between human and network F0 discrimination thresholds as the correlation between vectors of analogous data points. The network vector contained 60 F0 discrimination thresholds, one for each combination of phase and lowest harmonic number. To get a human vector with 60 analogous F0 discrimination thresholds, we (a) linearly interpolated the human data between lowest harmonic numbers and (b) assumed that F0 discrimination thresholds were constant for lowest harmonic numbers between 1 and 5 (supported by other published data[29,48]). We then computed the Pearson correlation coefficient between log-transformed vectors of human and network thresholds.

**Experiment B: pitch of alternating-phase harmonic complexes**. Experiment B reproduced the stimulus manipulation of Shackleton and Carlyon (1994) to test if our networks exhibited pitch-doubling for alternating-phase harmonic stimuli.

*Stimuli*. Stimuli consisted of consecutive harmonics (each presented at 50 dB SPL) summed together in alternating sine/cosine phase: odd-numbered harmonics in sine phase (0° offset between frequency components) and even-numbered harmonics in cosine phase (90° offset, such that components align at their peaks). As in Experiment A, these harmonic tones were bandpass-filtered and embedded in masking noise to control which harmonics were audible. The original study used pink noise and analog filters. Here, we used modified uniform masking noise and digital Butterworth filters (designed to approximate the original passbands). We generated stimuli with three different 4th-order Butterworth filters specified by their −3 dB passbands: 125–625 Hz ("low harmonics"), 1375–1875 Hz ("mid harmonics"), and 3900–5400 Hz ("high harmonics"). The exact harmonic numbers that are audible in each of these passbands depends on the F0. The original study used stimuli with F0s near 62.5, 125, and 250 Hz (sometimes offset by ±4% from the nominal F0 to avoid stereotyped responses). The 62.5 Hz condition was excluded here because the lowest F0 our networks could report was 80 Hz. We generated 354 stimuli with F0s near 125 Hz (120–130 Hz) and 250 Hz (240–260 Hz), in both cases uniformly sampled on a logarithmic scale, for each filter condition (2124 stimuli in total).

*Human experiment*. In the original experiment of Shackleton and Carlyon (1994), participants adjusted the F0 of a sine-phase control tone to match the pitch of a given alternating-phase test stimulus. The matched F0 provides a proxy for the perceived F0 for the test stimulus. The previously published human data were obtained from eight normal-hearing listeners who had a wide range of musical experience. Each participant made 18 pitch matches per condition.

*Model experiment*. To simulate the human paradigm in our model, we simply took the network's F0 prediction (within a 3-octave range centered at the stimulus F0) as the "perceived" F0 of the alternating-phase test stimulus. For each stimulus, we computed the ratio of the predicted F0 to the stimulus F0. Histograms of these frequency ratios (bin width = 2%) were generated for each of the six conditions (3

filter conditions × 2 nominal F0s). To simplify presentation, histograms are only shown for two conditions: "low harmonics" and "high harmonics", both with F0s near 125 Hz.

*Human-model comparison.* Shackleton and Carlyon (1994) constructed histograms from their pitch matching data, pooling responses across participants (144 pitch matches per histogram). We quantified the similarity between human and network responses by measuring correlations between human and network histograms for the same condition. Human histograms were first re-binned to have the same 2% bin width as network histograms. Pearson correlation coefficients were computed separately for each of the six conditions and then averaged across conditions to give a single number quantifying human-network similarity.

**Experiment C: pitch of frequency-shifted complexes.** Experiment C reproduced the stimulus manipulation of Moore and Moore (2003) to test if our networks exhibited pitch shifts for frequency-shifted complexes.

*Stimuli.* Stimuli were modifications of harmonic complex tones with consecutive harmonic frequencies in cosine phase. We imposed three different F0-dependent spectral envelopes–as described by Moore and Moore (2003)–on the stimuli. The first, which we termed the "low harmonics" spectral envelope, had a flat 3-harmonic-wide passband centered at the 5th harmonic. The second (termed "mid harmonics") had a flat 5-harmonic-wide passband centered at the 11th harmonic. The third (termed "high harmonics") had a flat 5-harmonic-wide passband centered at the 16th harmonic. All three of these spectral envelopes had sloping regions flanking the flat passband. Amplitudes (relative to the flat passband) at a given frequency $F$ in the sloping regions were always given by $(10^x - 1)/9$ where $x = 1 - |(F - F_e)/1.5F0|$ and $F_e$ is the edge of the flat region. The amplitude was set to zero for $x \le 0$.

For a given F0 and (fixed) spectral envelope, we made stimuli inharmonic by shifting every component frequency by a common offset in Hz specified as a percentage of the F0. As a concrete example, consider a stimulus with F0 = 100 Hz and the "low harmonics" spectral envelope. This stimulus contains nonzero energy at 200, 300, 400, 500, 600, and 700 Hz. Frequency-shifting this harmonic tone by +8% of the F0 results in an inharmonic tone with energy at 208, 308, 408, 508, 608, and 708 Hz. For each of the three spectral envelopes, we generated stimuli with frequency component shifts of +0, +4, +8, +12, +16, +20, and +24 %F0. For each combination of spectral envelope and frequency component shift, we generated stimuli with 3917 nominal F0s spaced log-uniformly between 80 and 480 Hz (83,391 stimuli in total). These stimuli are a superset of those used in the human experiment, which measured shifts for three F0s (100, 200, and 400 Hz) and four component shifts (+0, +8, +16, +24 %F0). As in the original study, stimuli were presented at overall levels of 70 dB SPL.

*Human experiment.* Moore and Moore (2003) used a pitch matching paradigm to allow listeners to report the perceived F0s for frequency-shifted complex tones. Five normal-hearing listeners (all musically trained) between the ages of 19 and 31 years old participated in the study. Each participant made 108 pitch matches. Moore and Moore (2003) reported quantitatively similar patterns of pitch shifts for the three F0s tested (100, 200, and 400 Hz). To simplify presentation, here we present their human data averaged across F0 conditions.

*Model experiment.* For the model experiment, we again took network F0 predictions for the 83,391 frequency-shifted complexes as the "perceived" F0s. F0 predictions were restricted to a one-octave range centered at the target F0 (the F0 of the stimulus before frequency-shifting). We summarize these values as shifts in the predicted F0, which are given by $(F0_{predicted} - F0_{target})/F0_{target}$. These shifts are reported as the median across all tested F0s and plotted as a function of component shift and spectral envelope. To simplify presentation, results are only shown for two spectral envelopes, "low harmonics" and "high harmonics".

*Human-model comparison.* We quantified the similarity between human and network pitch shifts as the Pearson correlation coefficient between vectors of analogous data points. The network vector contained 21 median shifts, one for each combination of spectral envelope and component shift. To obtain a human vector with 21 analogous pitch shifts, we linearly interpolated the human data between component shifts.

**Experiment D: pitch of complexes with individually mistuned harmonics.** Experiment D reproduced the stimulus manipulation of Moore et al. (1985) to test if our networks exhibited pitch shifts for complexes with individually mistuned harmonics.

*Stimuli.* Stimuli were modifications of harmonic complex tones containing 12 equal-amplitude harmonics (60 dB SPL per component) in sine phase. We generated such tones with F0s near 100 Hz, 200 Hz, and 400 Hz (178 F0s uniformly spaced on a logarithmic scale within ±4% of each nominal F0). Stimuli were then made inharmonic by shifting the frequency of a single component at a time. We applied +0, +1, +2, +3, +4, +6, and +8 % frequency shifts to each of the following harmonic numbers: 1, 2, 3, 4, 5, 6, and 12. In total there were 178 stimuli in each of the 147 conditions (3 nominal F0s × 7 component shifts × 7 harmonic numbers).

*Human experiment.* Moore et al. (1985) used a pitch matching paradigm in which participants adjusted the F0 of a comparison tone to match the perceived pitch of the complex with the mistuned harmonic. Three participants (all highly experienced in psychoacoustic tasks) completed the experiment. Participants each made ten pitch matches per condition tested. Humans were tested on 126 of the 147 conditions in the model experiment (3 nominal F0s × 7 component shifts × 6 harmonic numbers)—the conditions with a harmonic number of 12 were not included.

*Model experiment.* For the model experiment, we used the procedure described for Experiment C to measure shifts in the network's predicted F0 for all 26,166 stimuli. Shifts were averaged across similar F0s (within ±4% of the same nominal F0) and reported as a function of component shift and harmonic number. To simplify presentation, results are only shown for F0s near 200 Hz. Results were similar for F0s near 100 and 400 Hz.

*Human-model comparison.* We compared the network's pattern of pitch shifts to those averaged across the three participants from Moore et al. (1985). Human-model similarity was again quantified as the Pearson correlation coefficient between vectors of analogous data points. The network vector contained 147 mean shift values corresponding to the 147 conditions. Though Moore et al. (1985) did not report pitch shifts for the 12th harmonic, they explicitly stated they were unable to measure significant shifts when harmonics above the 6th were shifted. We thus inferred pitch shifts were always equal to zero for the 12th harmonic when compiling the vector of 147 analogous pitch shifts. We included this condition because some networks exhibited pitch shifts for high-numbered harmonics and we wanted our similarity metric to be sensitive to this deviation from human behavior.

**Experiment E: frequency discrimination with pure and transposed tones.** Experiment E measured network discrimination thresholds for pure tones and transposed tones as described by Oxenham et al. (2004).

*Stimuli.* Transposed tones were generated by multiplying a half-wave rectified low-frequency sinusoid (the "envelope") with a high-frequency sinusoid (the "carrier"). Before multiplication, the envelope was lowpass filtered (4th order Butterworth filter) with a cutoff frequency equal to 20% of the carrier frequency. To match the original study, we used carrier frequencies of 4000, 6350, and 10,080 Hz. For each carrier frequency, we generated 6144 transposed tones with envelope frequencies spaced uniformly on a logarithmic scale between 80 and 320 Hz. We also generated 6144 pure tones with frequencies spanning the same range. All stimuli were presented at 70 dB SPL and embedded in the same modified uniform masking noise as Experiment A. The original study embedded only the transposed tones in lowpass-filtered noise to mask distortion products. To ensure that the noise would not produce differences in the model's performance for the two types of stimuli, we included it for pure tones as well.

*Human experiment.* Oxenham et al. (2004) reported discrimination thresholds for these same four conditions (transposed tones with three different carrier frequencies + pure tones) at five reference frequencies between 55 and 320 Hz. Data were collected from four young (<30 years old) adult participants who had at least 1 h of training on the frequency discrimination task. Discrimination thresholds were based on three adaptive tracks per participant per condition.

*Model experiment.* The procedure for measuring network discrimination thresholds for pure tones was analogous to the one used in Experiment A. We first took network F0 predictions (within a one-octave range centered at the stimulus frequency) for all 6144 stimuli. We then simulated a two-alternative forced choice paradigm by making pairwise comparisons between predictions for stimuli with similar frequencies (within 2.7 semitones of five "reference frequencies" spaced log-uniformly between 80 and 320 Hz). For each pair of stimuli, we asked if the network correctly predicted a higher F0 for the stimulus with the higher frequency. From all trials at a given reference frequency, we constructed a psychometric function plotting the percentage of correct trials as a function of percent frequency difference between the two stimuli. Normal cumulative distribution functions were fit to each psychometric function and thresholds were defined as the percent frequency difference (capped at 100%) that yielded 70.7% correct. Each threshold was based on 233,586 pairwise discriminations made between 684 stimuli. The procedure for measuring thresholds with transposed tones was identical, except that the correct answer was determined by the envelope frequency rather than the carrier frequency. Thresholds were measured separately for transposed tones with different carrier frequencies. To simplify presentation, we show transposed tone thresholds averaged across carrier frequencies (results were similar for different carrier frequencies).

*Human-model comparison*. We quantified human-network similarity as the Pearson correlation coefficient between vectors of analogous log-transformed discrimination thresholds. Both vectors contained 20 discrimination thresholds corresponding to five reference frequencies × 4 stimulus classes (transposed tones with three different carrier frequencies + pure tones). Human thresholds were linearly interpolated to estimate thresholds at the same reference frequencies used for networks. This step was necessary because our networks were not trained to make F0 predictions below 80 Hz.

**Effect of stimulus level on frequency discrimination**. To investigate how phase locking in the periphery contributes to the level-robustness of pitch perception, we measured pure tone frequency discrimination thresholds from our networks as a function of stimulus level (Fig. 5d).

*Stimuli*. We generated pure tones at 6,144 frequencies spaced uniformly on a logarithmic scale between 200 and 800 Hz. Tones were embedded in the same modified uniform masking noise as Experiment A. The signal-to-noise ratio was fixed at 20 dB and the overall stimulus levels were varied between 10 and 100 dB SPL in increments of 10 dB.

*Human experiment*. Wier et al. (1977) reported frequency discrimination thresholds for pure tones in low-level broadband noise as a function of frequency and sensation level (i.e., the amount by which the stimulus is above its detection threshold). Thresholds were measured from four participants with at least 20 h of training on the frequency discrimination task. Participants completed four or five 2-down-1-up adaptive tracks of 100 trials per condition. Stimuli were presented at five different sensation levels: 5, 10, 20, 40, and 80 dB relative to masked thresholds in 0 dB spectrum level noise (broadband, lowpass-filtered at 10,000 Hz). We averaged the reported thresholds across four test frequencies (200, 400, 600, and 800 Hz) and re-plotted them as a function of sensation level in Fig. 5e.

*Model experiment*. We used the same procedure used in Experiments A and E to measure frequency discrimination thresholds. The simulated frequency discrimination experiment considered all possible pairings of stimuli with similar frequencies (within 2.7 semitones). Reported discrimination thresholds were pooled across all tested frequencies (200–800 Hz).

*Human-model comparison*. Because the human results were reported in terms of sensation level rather than SPL, we did not compute a quantitative measure of the match between model and human results, and instead plot the results side-by-side for qualitative comparison.

**Auditory nerve manipulations—overview**. The general procedure for investigating the dependence of network behavior on aspects of the auditory nerve representation was to (1) modify the auditory nerve model, (2) retrain networks (starting from a random initialization) on modified auditory nerve representations of the same natural sounds dataset, and (3) simulate psychophysical experiments on the trained networks using modified auditory nerve representations of the same test stimuli. We used this approach to investigate whether a biologically-constrained cochlea is necessary to obtain human-like pitch behavior and to evaluate the dependence of network pitch behavior on both temporal and "place" information in the auditory nerve representation. The only exception to this general procedure was in the experiment that tested the effect of flattening the excitation pattern, which was performed on networks that were trained on normal auditory nerve representations (see below).

**Replacing the hardwired cochlear model with learnable filters**. We replaced the hardwired auditory nerve model with a convolutional layer whose weights could be optimized alongside the rest of the DNN (Fig. 4). The convolutional layer consisted of 100 one-dimensional filter kernels (each with 801 taps) that operated directly on 32 kHz audio, applied using "valid" convolution. The audio input to the network was 75 ms in duration such that the valid output of convolution was 50 ms, as in the hardwired cochlear representation. Outputs from the 100 filters were stacked, half-wave rectified, and then resampled to 20 kHz, resulting in a first-layer representation with 100 "nerve fibers" and 1000 timesteps to match the size and temporal resolution of the hardwired cochlear representations. We separately trained the ten best network architectures from the original architecture search with this learnable "cochlear" layer. The best frequency of a learned filter was determined after training from the maximum value of its transfer function.

**Manipulating fine timing information in the auditory nerve**. We modified the upper frequency limit of phase locking in the auditory nerve by adjusting the cutoff frequency of the inner hair cell lowpass filter within the auditory nerve model. By default, the lowpass characteristics of the inner hair cell's membrane potential are modeled as a 7th order filter with a cutoff frequency of 3000 Hz[42]. We trained and tested networks with this cutoff frequency set to 50, 250, 1000, 3000, 6000, and 9000 Hz. In each of these cases, the sampling rate of the peripheral representation

used as input to the networks was 20 kHz so that spike-timing information would not be limited by the Nyquist frequency.

When the inner hair cell cutoff frequency is set to 50 Hz, virtually all temporal information in the short-duration stimuli we used was eliminated, leaving only place information (Fig. 5a). To control for the possibility that the performance characteristics of networks trained on such representations could be limited by the number of model nerve fibers (set to 100 for most of our experiments), we repeated this manipulation with 1000 auditory nerve fibers (characteristic frequencies again spaced uniformly on an ERB-number scale between 125 Hz and 14,000 Hz). To keep the network architecture constant, we reduced the sampling rate to 2 kHz (which for the 50 Hz hair cell cutoff preserved all stimulus-related information), yielding peripheral representations that were 1000-fiber by 100-timestep arrays of instantaneous firing rates. We then simply transposed the nerve fiber and time dimensions so that networks still operated on 100-by-1000 inputs, allowing us to use the same network architectures as in all other training conditions. Note that by transposing the input representation, we effectively changed the orientation of the convolutional filter kernels. Kernels that were previously long in the time dimension and short in the nerve fiber dimension became short in the time dimension and long in the nerve fiber dimension. We saw this as desirable as it allowed us to rule out the additional possibility that the performance characteristics of networks with lower limits of phase locking were due to convolutional kernel shapes that were optimized for input representations with high temporal fidelity and thus perhaps less suited for extracting place information (which requires pooling information across nerve fibers).

To more closely examine how the performance with degraded phase locking (i.e., the 50 Hz inner hair cell cutoff frequency condition) might be limited by the number of model nerve fibers, we also generated peripheral representations with either 100, 250, or 500 nerve fibers (with characteristics frequencies uniformly spaced on an ERB-number scale between 125 Hz and 14000 Hz in each case). To keep the network's input size fixed at 100-by-1000 (necessary to use the same network architecture), we transposed the input array, again using 100 timesteps instead of 1000 (sampled at 2 kHz), and upsampled the frequency (nerve fiber) dimension to 1000 via linear interpolation. In this way the input dimensionality was preserved across conditions even though the information was limited by the original number of nerve fibers. Median %F0 error on the validation set and discrimination thresholds and were measured for networks trained and tested with each of these peripheral representations (Supplementary Fig. 4).

**Eliminating place cues by flattening the excitation pattern**. To test if our trained networks made use of peaks and valleys in the time-averaged excitation pattern (which provide "place" cues to F0), we tested networks on nerve representations with artificially flattened excitation patterns. Nerve representations were flattened by separately scaling each frequency channel of the nerve representation to have the same time-averaged response. Each row (nerve fiber) of the nerve representation was divided by its time-averaged firing rate and multiplied by the mean firing rate across all rows, yielding an excitation-flattened nerve representation with the same mean firing rate as the original. This manipulation was separately applied to each psychophysical stimulus.

**Manipulating cochlear filter bandwidths**. Cochlear filter bandwidths in the auditory nerve model were set based on estimates of human frequency tuning from otoacoustic and behavioral experiments[53]. We modified the frequency tuning to be two times narrower and two times broader than these human estimates by scaling the filter bandwidths by 0.5 and 2.0, respectively.

To investigate the importance of the frequency scaling found in the cochlea, we also generated a peripheral representation with linearly spaced cochlear filters. The characteristic frequencies of 100 model nerve fibers were linearly spaced between 125 Hz and 8125 Hz and the 10-dB-down bandwidth of each cochlear filter was set to 80 Hz. This bandwidth (which is approximately equal to that of a "human" model fiber with 400 Hz characteristic frequency) was chosen to be as narrow as possible without introducing frequency "gaps" between adjacent cochlear filters.

To verify that these manipulations had the anticipated effects, we measured tuning curves (detection thresholds as a function of frequency) for simulated nerve fibers with characteristic frequencies of 250, 500, 1000, 2000, 4000 Hz (Fig. 6a, b). Mean firing rate responses were computed for each fiber to 50 ms pure tones with frequencies between 125 Hz and 8000 Hz. Thresholds were defined as the minimum sound level required to increase the fiber's mean firing rate response 10% above its spontaneous rate (i.e., dB SPL required for 77 spike/s).

**Sound statistics manipulations—overview**. The general procedure for investigating the dependence of network behavior on sound statistics was to (1) modify the sounds in training dataset, (2) retrain networks (starting from a random initialization) on auditory nerve representations of the modified training dataset, and (3) simulate psychophysical experiments on trained networks, always using the same test stimuli.

**Training on filtered natural sounds**. We generated lowpass and highpass versions of our natural sounds training dataset by applying randomly-generated lowpass or highpass Butterworth filters to every speech and instrument sound excerpt. For the

lowpass-filtered dataset, 3-dB-down filter cutoff frequencies were drawn uniformly on a logarithmic scale between 500 and 5000 Hz. For the highpass-filtered dataset, cutoff frequencies were drawn uniformly from a logarithmic scale between 1000 and 10,000 Hz. The order of each filter was drawn uniformly from one to five and all filters were applied twice, once forward and once backwards, to eliminate phase shifts. Filtered speech and instrument sounds were then combined with the unmodified background noise signals used in the original dataset (SNRs drawn uniformly from −10 to +10 dB).

**Training on spectrally matched and anti-matched synthetic tones.** To investigate the extent to which network pitch behavior could be explained by low-order spectral statistics of our natural sounds dataset, we generated a dataset of 2.1 million synthetic stimuli with spectral statistics matched to those measured from our primary dataset. STRAIGHT[75] was used to measure the spectral envelope (by averaging the estimated filter spectrogram across time) of every speech and instrument sound in our dataset. We then measured the mean and covariance of the first 13 Mel-frequency cepstral coefficients (MFCCs), defining a multivariate Gaussian. We sampled new spectral envelopes from this distribution by drawing MFCC coefficients and inverting them to produce a spectral envelope. These envelopes were imposed (via multiplication in the frequency domain) on harmonic complex tones with F0s sampled to uniformly populate the 700 log-spaced F0 bins in the network's classification layer. Before envelope imposition, tones initially contained all harmonics up to 16 kHz in cosine phase, with equal amplitudes.

To generate a synthetic dataset with spectral statistics that deviate considerably from those measured from our primary dataset, we simply multiplied the mean of the fitted multivariate Gaussian (a vector of 13 MFCCs) by negative one, which inverts the mean spectral envelope. Spectral envelopes sampled from the distribution defined by the negated mean (and unaltered covariance matrix) were imposed on 2.1 million harmonic complex tones to generate an "anti-matched" synthetic tones dataset.

Both the matched and anti-matched synthetic tones were embedded in synthetic noise spectrally matched to the background noise in our primary natural sounds dataset. The procedure for synthesizing spectrally-matched noise was analogous to the one used to generate spectrally-matched tones, except that we estimated the spectral envelope using the power spectrum. We measured the power spectrum of every background noise clip in our primary dataset, computed the mean and covariance of the first 13 MFCCs, and imposed spectral envelopes sampled from the resulting multivariate Gaussian on white noise via multiplication in the frequency domain. Synthetic tones and noise were combined with SNRs drawn uniformly from −10 to +10 dB, and overall stimulus presentation levels were drawn uniformly from 30 to 90 dB SPL.

**Training on speech and music separately.** We generated speech-only and music-only training datasets by selectively sampling from the same source libraries used to populate the combined dataset. Due to the lack of speech clips in our source libraries with high F0s, we decided to limit both datasets to F0s between 80 and 450 Hz (spanning 480 of 700 F0 bins). This ensured that differences between networks trained on speech or music would not be due to differences in the F0 range. The composition of the speech-only dataset was:

- F0 bins between 80 Hz and 320 Hz
  100% adult speech (2000 SWC and 1000 WSJ clips per bin)

- F0 bins between 320 Hz and 450 Hz
  100% child speech (1500 CSLU and 1500 CMU clips per bin)

The composition of our music-only dataset was:

- F0 bins between 80 Hz and 450 Hz
  100% instrumental music (2000 NSynth and 1000 RWC clips per bin)

Stimuli in both datasets were added to background noise clips sampled from the same sources used for the combined dataset (SNRs drawn uniformly from −10 to +10 dB).

**Training on natural sounds with reduced background noise.** To train networks in a low-noise environment, we regenerated our natural sounds training dataset with SNRs drawn uniformly from +10 to +30 dB rather than −10 to +10 dB. For the noiseless case, we entirely omitted the addition of background noise to the speech and instruments sounds before training. To ensure F0 discrimination thresholds measured from networks trained with reduced background noise would not be limited by masking noise in the psychophysical stimuli, we evaluated these networks on noiseless versions of the psychophysical stimuli (Experiments A and E). The amplitudes of harmonics that were masked by noise in the original stimuli (i.e., harmonics inaudible to human listeners in the original studies) were set to zero in the noiseless stimuli. When these networks were evaluated on

psychophysical stimuli that did include masking noise, F0 discrimination behavior was qualitatively similar, but absolute thresholds were elevated relative to networks that were trained on the −10 to +10 dB SNR dataset.

**Network neurophysiology.** We simulated electrophysiological recordings and functional imaging experiments on our trained networks by examining the internal activations of networks in response to stimuli. We treated units in the network layers as model "neurons" and looked at their tuning using their average activations across time to different stimuli. We measured tuning properties using equal-amplitude sine-phase harmonic complex tones (45 dB SPL per harmonic component) in threshold equalizing noise (10 dB SPL per ERB). We used a total of 11,520 stimuli: five unique harmonic compositions (pure tones or successive harmonics 1–9, 2–10, 4–12, or 6–14) × 2304 unique F0s (logarithmically-spaced between 80 and 640 Hz). For each unit, we constructed an F0 tuning curve for each harmonic composition by averaging activations to stimuli within the same F0 classification bin (i.e., within 1/16 semitone bins). Tuning curves were normalized separately for each unit by dividing by the unit's maximum response across the full stimulus set. Units that produced a response of zero to all of the test stimuli were excluded from analysis (< 1% of units).

As a measure of the strength of F0 tuning in a single model unit, we computed the mean Pearson correlation coefficient between the pure tone (frequency) tuning curve and each of the complex tone tuning curves. A perfectly F0-tuned unit should selectively respond to pure and complex tones of a preferred F0 independent of harmonic composition (Fig. 9a), yielding a high mean correlation coefficient. We measured the F0 tuning correlation of each unit in each layer (all ReLU activations following convolutional layers and fully connected layers) of the ten best-performing network architectures. The F0 tuning strength of a network layer was computed by averaging this metric across all units in the layer.

To test if the observed F0 tuning depended on the harmonicity of the stimuli, we repeated this analysis with complex tones made inharmonic by randomly shifting component frequencies with a fixed jitter pattern[20]. The jitter pattern allowed for each individual component (after the F0 component) to be shifted by up to ±50% of the F0. Jitter values for each component were drawn uniformly from −50% to +50% with rejection sampling to ensure adjacent components were separated by at least 30 Hz to minimize salient differences in beating. Each component's frequency was the original harmonic's frequency plus the jitter value multiplied by the nominal F0. Like harmonic tones, inharmonic tones generated in this way have frequency components that increase in spacing as the nominal F0 is increased, but unlike harmonic tones they lack a fundamental frequency in the range of audible pitch (i.e., above ~30 Hz[86]). Empirically, human perceptual signatures of F0-based pitch are disrupted by inharmonicity[20,55,57], making it a way to distinguish human-like representations of F0 from coarser representations of frequency spacing. The same jitter pattern (i.e., the same mapping of nominal harmonic numbers to jitter value) was applied to all stimuli regardless of F0 and harmonic composition. As in the F0 tuning analysis with harmonic tones, we measured the average strength of nominal F0 tuning for each layer in the ten networks. To ensure results were not unduly biased by a single random jitter pattern, the analysis was repeated five times with different random seeds. F0 tuning summary metrics reported in Fig. 9c are averaged across these five random seeds.

We measured population responses as a function of lowest harmonic number (Fig. 9d) using a superset of the harmonic complex tones used to measure F0 tuning. The dataset was expanded to include all complexes containing 9 successive harmonics, with lowest harmonic numbers 1–15 (e.g., 1–9, 2–10, 3–11, … 15–26). We first identified the best F0 (i.e., the F0 producing the largest normalized mean response across all lowest harmonic numbers) for each of the 700 units in each network's final fully connected layer. We then constructed lowest-harmonic-number tuning curves by taking responses to stimuli at the best F0 with lowest harmonic numbers 1–15. These tuning curves were averaged across units to give the population response.

We qualitatively compared the network's population response to those of pitch-selective neurons in marmoset auditory cortex[28] and pitch-selective voxels in human auditory cortex[29]. Bendor and Wang used single-unit electrophysiology to measure the spiking rates of 50 pitch-selective neurons (from three marmosets) in response to complexes containing 3–9 consecutive harmonics in either cosine or Schroeder phase. Recordings were repeated a minimum of ten times per stimulus condition. Norman-Haignere and colleagues measured fMRI responses to bandpass-filtered sine-phase harmonic complex tones and frequency-matched noise. The 12 participants (four male, eight female, ages 21–28) were non-musicians with normal hearing. Pitch-selective voxels were defined as those whose responses were larger for complex tones than for frequency-matched noise. We re-plotted data extracted from figures in both published studies.

**Statistics—analysis of human data.** Human data were scanned from original figures or provided by the authors of the original papers. We did not have access to data from individual human participants, and so did not plot error bars on the graphs of human results.

**Statistics—analysis of human-model comparison metrics.** Human-model comparison metrics were computed separately for each psychophysical experiment

(as described in the above sections on each experiment and its analysis) and for each of the 400 networks trained in our architecture search. To test if networks with better performance on the F0 estimation training task produce better matches to human psychophysical behavior, we computed the Pearson correlation between validation set accuracies and human-model comparison metrics for each experiment (Fig. 3, right-most column).

To test if "deep" networks (defined here as networks with more than one convolutional layer) tended to produce better performance on the training task than the networks with just one convolutional layer (Supplementary Fig. 3), we performed a Wilcoxon rank-sum test comparing the validation set accuracies of the 54 single-convolutional-layer networks to those of the other 346 networks. To test if the "deep" networks tended to produce better matches to human behavior we performed a Wilcoxon rank-sum test comparing the human-model similarity metrics of the 54 single-convolutional-layer networks to those of the other 346 networks. To obtain a single human-model similarity score per network for this test, we pooled metrics across the five main psychophysical experiments. This was accomplished by first rank ordering the human-model similarity metrics of all networks within experiments and then averaging ranks across experiments.

These human-model similarity metrics were used to analyze two other experiments (in all others we used more fine-grained analysis of best thresholds and transition points, described below). To assess the statistical significance of changes in human-model similarity when networks were optimized with a learned "cochlea" (Fig. 4) or in the absence of background noise, we compared metrics measured from ten networks trained per condition. The networks had ten different architectures, corresponding to the ten best-performing architectures identified in our search (Supplementary Table 1). We performed two-sample t-tests (each sample containing results from the ten independently trained networks) to compare human-model comparison metrics between training conditions. Effect sizes were quantified as Cohen's d and reported for all such tests that indicated statistically significant differences. Because human-model similarity metrics were bounded between −1 and 1, we passed the metrics through an inverse normal cumulative distribution function before performing t-tests. All t-tests and rank-sum tests were two-sided.

**Statistics—analysis of best thresholds and transition points**. One of the key signatures of human pitch perception is that listeners are very good at making fine F0 discriminations (thresholds typically below 1%) if and only if stimuli contain low-numbered harmonics. F0 discrimination thresholds increase by an order of magnitude for stimuli containing only higher-numbered harmonics[43,48]. To assess the effect of altered cochlear input or training sound statistics (Figs. 5–7), we thus focused on two measures: first, the absolute F0 discrimination acuity of our model when all low-numbered harmonics were present ("best threshold"), and second, the harmonic number at which discrimination ability transitioned from good to poor ("transition point"). In each case, we used two-sample t-tests, comparing either the F0 discrimination thresholds (log-transformed) for tones containing the first harmonic, or the lowest harmonic number where thresholds first exceeded 1%. In each case we compared results for networks with different auditory nerve models or training sets. To quantify effect sizes, Cohen's d is reported for all two-sample t-tests that indicated statistically significant differences.

**Statistics—ANOVAs on discrimination thresholds**. We performed analyses of variance (ANOVAs) on log-transformed F0 discrimination thresholds to help satisfy the assumptions of equal variance and normality (normality was evaluated by eye). Mixed model ANOVAs were performed with training conditions (peripheral model and training set manipulations) as between-subject factors and psychophysical stimulus parameters (lowest harmonic number and stimulus presentation level) as within-subject factors. The specific pairings of these different factors were: stimulus presentation level vs. auditory nerve phase locking cutoff (Fig. 5e), lowest harmonic number vs. training set spectral statistics (Fig. 7c), and lowest harmonic number vs. training set noise level (Fig. 8). We also performed a repeated-measures ANOVA to test for a main effect of lowest harmonic number on population responses in the network's last layer (Fig. 9c). F-statistics, p-values, and $\eta^2_{partial}$ are reported for main effects and interactions of interest. Greenhouse-Geisser corrections were applied in all cases where Mauchly's test indicated the assumption of sphericity had been violated.

**Reporting summary**. Further information on research design is available in the Nature Research Reporting Summary linked to this article.

## Data availability

The Wall Street Journal (LDC93S6A), CMU Kids Corpus (LDC97S63), and CSLU Kids Speech (LDC2007S18) audio datasets used in this study are available from the Linguistic Data Consortium (https://www.ldc.upenn.edu). The Spoken Wikipedia Corpora audio dataset is available at https://nats.gitlab.io/swc/. The RWC Music Database is available at https://staff.aist.go.jp/m.goto/RWC-MDB/. The pitch datasets we compiled from these publicly available corpora, along with the psychophysical test stimulus sets, are available at: https://github.com/msaddler/pitchnet. Source data are provided with this paper.

## Code availability

Source code for the Bruce et al. (2018) auditory nerve model is available at https://www.ece.mcmaster.ca/~ibruce/zbcANmodel/zbcANmodel.htm. We developed a Python wrapper around the model, which supports flexible manipulation of cochlear filter bandwidths and the upper limit of phase locking. This wrapper is available at https://github.com/msaddler/bez2018model. Code to implement and analyze our deep neural network pitch models (including trained network weights) is available at https://github.com/msaddler/pitchnet.

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

## Acknowledgements
We thank Alex Durango and Jenelle Feather for providing the AudioSet background noise stimuli, Jenelle Feather and Andrew Francl for contributions to a shared codebase used in this project, John Cohn for assistance with computing resources, Josh Bernstein, Andrew Oxenham, Trevor Shackleton, and Bob Carlyon for sharing human psychophysics data and stimuli, Sam Norman-Haignere and the McDermott lab for helpful feedback on the manuscript, and Ian Bruce, Laurel Carney, Bertrand Delgutte, Oded Barzelay, Brian Moore, and Hideki Kawahara for helpful discussions. Work supported by NSF grant BCS-1634050 and NIH grant R01DC017970.

## Author contributions
All authors conceived the project and designed the experiments. R.G. ran and analyzed pilot versions of the experiments. M.R.S. ran the experiments described in the paper, analyzed the data, and made the figures. M.R.S. and J.H.M. drafted the manuscript. All authors edited the manuscript.

## Competing interests
The authors declare no competing interests.
