## [Peer Review File · Nature Communications]

Deep neural network models reveal interplay of peripheral coding and stimulus statistics in pitch perceptionREVIEWER COMMENTS

Reviewer #1 (Remarks to the Author):

In this exciting manuscript, the authors examine a long-standing problem in auditory neuroscience—the underlying mechanisms of pitch perception. Where earlier computational models and experimental data have not been able to fully address this question, the authors use a deep neural network architecture to achieve fundamental frequency discrimination with a performance similar to that of a human listener, and perturb their model in a multitude of ways to investigate how pitch is processed. While we don't know whether this particular deep neural network works using the same strategies as the human brain, the authors are able to accurately replicate many of the psychophysical pitch phenomena observed in humans (e.g. sin/rand phase f0 discrimination differences). Furthermore, this approach provides substantial flexibility in testing changes to the model not possible in nature (e.g. linearly spaced frequency tuning, training without background noise). Overall a well written and insightful manuscript. Only a few minor comments.

Figure 4—superior phase locking (6kHz and 9 kHz) do not improve F0 discrimination thresholds at low harmonic numbers, but actually seem to do worse at high harmonic numbers. Why do you think this is the case?

Figure 5—Thresholds are worse for higher harmonic numbers for narrow and wide filters, compared to intermediate (human filter BW). Can you speculate on the cause of this observation?

Figure 5- To follow up on the previous comment, other species can have very different auditory filters, and different pitch discrimination thresholds (compared to humans). However, these deep neural networks seem very robust to filter changes (especially at low harmonic numbers). Can these DNNs explain why non-human species have substantially worse pitch discrimination thresholds?

Figure 8- while tuning gets sharper in deep layers of the network, its worth noting that it doesn't look much like neural tuning curves (which have more of a gaussian shape), and the background level of activity (relative to the peak mean activation) is the biggest factor changing. However, looking at F0 tuning to a complex tone alone may not elucidate the processing happening between layers. At the very least you should compare pure tone and missing fundamental tuning. There is evidence of this occurring potentially in example units 3642 and unit 4605 (Fig 8A), but the prevalence of this, and how early it occurs is important to show.

Reviewer #2 (Remarks to the Author):

Review of “Deep neural network models reveal interplay of peripheral coding and stimulus statistics in pitch perception” by Saddler, Gonzales and McDermott :

This paper reports the results of what I believe is an important step forward in hearing sciences, advancing significantly our knowledge on pitch-perception auditory mechanisms in humans (pitch perception being a capacity playing a key role in speech and music recognition). More broadly, this paper also advances significantly our understanding of the complex interactions between environmental and biological constraints in perception.

The study is based on an original computational approach making elegant use of a classical model of the human cochlea (our sensory receptor for the auditory modality) together with recent advances in machine-learning techniques. This approach allows a series of experimental tests that cannot be conducted in real human listeners : i) manipulating systematically spectral and temporal coding fidelity at the peripheral (cochlear) level; ii) training the central architecture of the pitch-discrimination system with speech, music, or synthetic tones with or without altered characteristics). In short, this study makes two extremely valuable contributions for a journal like Nature Communication: not only does it provide a clear answer to a debate on the role of peripheral sensory mechanisms that has been revolving for more than half a century (ie, whether pitch perception is based on temporal vs spectral/spatial neural cues available at the sensory-receptor level), but it also provides clear answers to another fundamental issue in sensory sciences (ie, whether characteristics of natural stimuli shape central sensory mechanisms). In addition to that, this study makes surprising discoveries on intriguing aspects of pitch perception, such as the “auditory perception of the missing fundamental”, by pointing to a role of hearing in noise. I entirely agree with the authors that the current approach based on an optimization procedure is useful to provide important insights into human behaviour, because the ideal observers are quite difficult - if not impossible to figure out - for the (relatively complex) perceptual tasks tested here. The fact that the best-performing deep neural networks mimicked both qualitatively and quantitatively human data relatively well is another important theoretical finding of this study, revealing that humans make a near-optimal use of the low-level sensory cues conveyed by their cochleas.

The manuscript is extremely well written and made easy to follow for a wide audience including of course psychophysicists, neuroscientists, modellers, but also physicists, computer scientists and clinicians with some background in neurosciences and experimental audiology (eg, ENT clinicians and audiologists). All figures are well designed and present the main findings in a straightforward way. The vast literature on the present topic is well reviewed. The method is sound and the paper provides enough detail for the work to be reproduced; the data are well analysed and well discussed. For all these reasons, I believe that this manuscript deserves being published in Nature Communication. For the same reasons, this paper certainly merits being cited in future hearing-science manuals. I have only six specific but minor comments to make on this study that the authors may wish to take into account to improve their manuscript.

Specific comments:

1-Dependence on low-order harmonics (experiment A; cf. Fig 2A): The authors acknowledge several times that the inflection point on the curve showing simulated discrimination data was lower than in real human listeners. I suspect that this reflects characteristics of the model of peripheral auditory

system used in here. The authors should elaborate more on this relative failure of the modelling approach, and attempt to explain why the inflection moves to the 10-15th harmonic in real listeners.

2-Perception of transposed tones (experiment E; cf. Fig 2E; Figure 3E): this now “classical” result of the literature suggesting that a correct spatial (tonotopic representation) is requested for normal pitch perception is the only situation where the accuracy of model predictions are not related to the optimization process. Here again, the authors should elaborate on the reasons why this task does not follow the same trends as the other tasks.

3-I was amazed by the relatively modest effects of altering cochlear frequency selectivity on the model’s performance. The results are convincing and this clearly is a major finding of the present study as most members of the hearing science community would assume that cochlear frequency selectivity strongly constrains pitch discrimination. As to the demonstration of the role of neural phase locking (ie, temporal fine structure) cues in pitch perception, I was especially fascinated by the fact that an upper limit of neural phase locking at 3kHz and above is required to reproduce the invariance of pitch perception to level shown by real human listeners. This as such is another key finding of the present study, highlighting the reason why neural phase locking is required to obtain robust perception for normal-hearing listeners. This finding has also important audiological implications for the understanding of the detrimental effects of cochlear damage (eg, caused by ageing or noise exposure) on auditory perception for hearing-impaired people. Recent neurophysiological studies indicate that cochlear damage causes a mismatch between neural temporal-fine structure information and the place on the basilar membrane that would “normally” respond to that information (Henry et al., 2013, 2016, 2019). I guess that this impact of such cochlear alterations on pitch perception may be simulated with the present approach by modifying certain aspects of the cochlear model (eg, by modifying the tip/tail component of cochlear-filters tuning curves). Such data would certainly trigger a huge interest in clinicians and hearing-aid manufacturers.

4-Ability to extract F0 information from high-number harmonics and perception of missing fundamental: A very intriguing result suggests that this ability results partly from exposure to noise during training (eg, during auditory development). Some elaborations, or even speculations, would be welcome here because noise is expected to corrupt any kind of (temporal and/or spectral) cues rather than helping the discrimination process. One may speculate about a potential “filling in” phenomenon caused by noise. For instance, could it be the case that background noise elicits (poor) temporal cues at the output of low center-frequency (CF) cochlear channels tuned at or close the (missing) F0 that may assist the optimization process? In other words, at low CF channels required during the training phase (to test this, these channels may be turned off)?

5-Network physiology: Simulated electrophysiology experiments on the best-performing networks showed that units in the deepest layers were selectively and sharply tuned to F0, consistent with electrophysiological data for non-human animals and brain-imaging data for humans. This is a nice result that certainly deserves further characterization. How does such a tuning emerge in the processing architecture? What are the low-level mechanisms allowing for the conversion of peripheral (time, place, time and place) information into a “rate-place” code for pitch ?

6-Future directions: By showing that pitch discrimination depends both on the information available at the output of the peripheral auditory system and on the environmental constraints in which the auditory system is optimized, this study opens an exciting path that may renew the study of auditory development (with and without sensorineural hearing loss; eg understanding the variable outcome of paediatric cochlear implantation) or auditory acclimatization to hearing aids for adults. The

authors may wish to make a short note pointing in this direction (this is already suggested on line 653).

End.

References:

Heinz MG, Henry KS (2013) Modeling disrupted tonotopicity of temporal coding following sensorineural hearing loss. *Proc Mtgs Acoust* 19:050177.

Henry KS, Kale S, Heinz MG (2016) Distorted tonotopic coding of temporal envelope and fine structure with noise-induced hearing loss. *J Neurosci* 36:2227–2237.

Henry KS, Sayles M, Hickox, ME, HeinzMG (2019) Divergent Auditory Nerve Encoding Deficits Between Two Common Etiologies of Sensorineural Hearing Loss. *J Neurosci*, 39(35):6879–6887

Reviewer #3 (Remarks to the Author):

Review of “Deep neural network models reveal interplay of peripheral coding and stimulus statistics in pitch perception” by Saddler et al.

This article proposes a computational approach towards teasing out the components of pitch perception attributable to ear physiology as opposed to the environment.

The authors use an approach that has become common in computational neuroscience, namely to train deep neural networks for a task, pitch discrimination in this case, and to utilize the network to gain normative insights into how the brain achieves the task. The authors trained deep neural networks for pitch estimation using natural sounds pre-processed with a cochlear model and found that the networks exhibit human-like behavior. The authors demonstrate that altering the statistics of the sound used to train the networks alters the behavior of the networks, making them less human-like for “un-natural” sounds, which, they suggest, points to an environmental component to pitch perception.

The novelty of this work seems to stem, not on the use of deep learning per se, but on its use in conjunction with inputs of different statistics to arrive at the conclusion of an environmental component to pitch perception.

I appreciate the thoroughness with which the authors describe their deep learning experiments, and the experiments performed to compare the performance of 'the model' to humans. I found the fact that 'the model' exhibits human-like behavior interesting, as as the authors' characterization of network psychophysics. Below, I make a few comments/suggestions to the authors that I believe can help to improve the manuscript and, possibly, lead to interesting computational findings.

1 Can the architecture learn the cochlear pre-processing module from data?

By using a cochlear model as a pre-processing, I understand the authors' motivation to impose physiological constraints on the architecture. Personally, I would find the authors' finding much more interesting if the networks could learn this pre-processing stage. As far as I understand, from a deep learning perspective, this stage has a simple interpretation that one could specify as trainable layer.

In the section "Human-like behavior is less dependent on cochlear filter bandwidths", the authors say that "Linearly-spaced cochlear filters also yielded best thresholds that were not significantly different from those for normal human tuning...". The Fourier basis represents the simplest example of a linearly-spaced filter bank (you can think of the filters as either complex exponentials or sines/cosines). In the time domain, we can interpret filtering operations as convolutions, suggesting that the authors can replace the cochlear pre-processing model with a convolutional layer with one-dimensional convolutional filters (100 of them for the authors), followed by a Fourier transform and a magnitude nonlinearity

to obtain. Applying this in successive windows of input sounds would yield a 100 x 1000 representation of the same shape as that obtained by the cochlear module. I would suggest to the authors that they train the filters from this layer. In my opinion, analyzing the filters from this layer and comparing them to cochlear processing could reveal differences that lead to (likely) better pitch discrimination than with the cochlear module. I also suspect that training this layer for sound from different statistics may close the gap the authors see between speech and music performance. In other words, I suspect that the difference the authors see for different sound statistics, which I personally do not find surprising, comes from the cochlear-processing step. I would find it interesting that the architecture learns a filter bank tailored to statistics of the input sounds.

Note: What I suggest may involve neural nets with complex values, depending on implementation. There exists a rich literature on the topic.

2 Network averaging and Bayesian deep learning/ensemble models

By averaging networks to obtain ‘the model’, the authors essentially form deep ensembles [1]. These have garnered recent interest, particularly in the context of Bayesian deep learning [2]. I would encourage the authors to cite this literature, put ‘the model’ explicitly in the context of this literature, and to think about ways in which they currently/could leverage Bayesian deep learning (perhaps without knowing it) in their work. In my opinion, doing so could help to raise awareness, within the neuroscience community, as to the issues of uncertainty quantification in deep learning.

References

- [1] Lakshminarayanan, B., Pritzel, A., & Blundell, C. “Simple and scalable predictive uncertainty estimation using deep ensembles”. In *Advances in Neural Information Processing Systems*, pp. 6402-6413, 2017.
- [2] Wilson, Andrew Gordon. ”The case for Bayesian deep learning.” arXiv preprint arXiv:2001.10995 (2020).

Please note that all line numbers are taken from the pdf for review that has figures embedded in the text for ease of reading. We have also included a version with tracked changes to make it easy to see the revisions.

Reviewer #1 (Remarks to the Author)

In this exciting manuscript, the authors examine a long-standing problem in auditory neuroscience—the underlying mechanisms of pitch perception. Where earlier computational models and experimental data have not been able to fully address this question, the authors use a deep neural network architecture to achieve fundamental frequency discrimination with a performance similar to that of a human listener, and perturb their model in a multitude of ways to investigate how pitch is processed. While we don't know whether this particular deep neural network works using the same strategies as the human brain, the authors are able to accurately replicate many of the psychophysical pitch phenomena observed in humans (e.g. sin/rand phase f0 discrimination differences). Furthermore, this approach provides substantial flexibility in testing changes to the model not possible in nature (e.g. linearly spaced frequency tuning, training without background noise). Overall a well written and insightful manuscript. Only a few minor comments.

Thank you!

Figure 4—superior phase locking (6kHz and 9 kHz) do not improve F0 discrimination thresholds at low harmonic numbers, but actually seem to do worse at high harmonic numbers. Why do you think this is the case?

For networks with IHC cutoff frequencies between 320 Hz and 9000 Hz, there is a trend that lower phase-locking cutoffs produce slightly better F0 discrimination thresholds at high harmonic numbers. When the peripheral representation contains less high frequency phase-locked spike timing information, there is a greater incentive for networks to use envelope timing cues to extract F0 information (because nerve fibers have a reduced ability to phase-lock to the fine structure). This likely biases the learned strategy of the networks, resulting in better thresholds for stimuli containing only high-numbered harmonics (for which envelope F0 cues are particularly important). When the peripheral representation contains superior phase-locking (6 kHz and 9 kHz IHC cutoffs), the reverse likely happens. The networks are incentivized to rely more on fine structure spike-timing (because it enables better F0 estimation on the speech/instrument dataset), which is less useful for stimuli containing only high harmonics.

We have added discussion of this issue to the relevant section of the Results:

“Discrimination thresholds for high-numbered harmonics were in fact slightly worse for increased cutoff frequencies. One explanation is that increasing the model’s access to fine timing information biases the learned strategy to rely more on this information, which is less useful for determining the F0 of stimuli containing only high-numbered harmonics.” (lines 347-351)

Figure 5—Thresholds are worse for higher harmonic numbers for narrow and wide filters, compared to intermediate (human filter BW). Can you speculate on the cause of this observation?

Somewhat poorer thresholds at higher harmonic numbers can be expected for networks trained with narrower cochlear filters, because narrower filters result in less salient envelope F0 cues (from the beating of adjacent harmonics). However, we are puzzled why thresholds for the broader filter networks are also poorer.

We ruled out one possible explanation for the worse thresholds with altered cochlear filter bandwidths: overfitting of the architecture. Since we performed the architecture search with human filter bandwidths, it is plausible that the best network architectures are overfit to the human filter bandwidths. However, we found no significant differences between the validation set accuracies when we made the filters broader or narrower, suggesting this was not the case.

Though we do not have an explanation for the poorer thresholds at high harmonics, we emphasize that the effect is not obviously consequential. For stimuli containing only high harmonics, human pitch perception is generally poor (thresholds are about an order of magnitude worse than they are for low harmonics) and all of the models exhibit what would be considered poor performance in this regime, and are thus all generally consistent with human perception for such stimuli.

We have clarified this issue in the revised results section:

“All three models with altered cochlear filter bandwidths produced worse thresholds for stimuli containing only high-numbered harmonics (Fig. 6D). While this is expected for the narrower and linearly-spaced conditions (smaller bandwidths result in reduced envelope cues from beating of adjacent harmonics), we do not have an explanation for why networks with broader filters also produced poorer thresholds. One possibility that we ruled out is overfitting of the network architectures to the human cochlear filter bandwidths; validation set accuracies were no worse with broader filters ($t(18)=0.66$, $p=0.52$; two-sample t -test). However, we note that all of the models exhibit what would be considered poor performance for stimuli containing only high harmonics (thresholds are at least an order of magnitude worse than they are for low harmonics), and are thus all generally consistent with human perception in this regime.” (lines 430-440)

Figure 5- To follow up on the previous comment, other species can have very different auditory filters, and different pitch discrimination thresholds (compared to humans). However, these deep neural networks seem very robust to filter changes (especially at low harmonic numbers). Can these DNNs explain why non-human species have substantially worse pitch discrimination thresholds?

Our results suggest that differences between species are more likely driven by differences in auditory environment and/or the ecological need for an optimized F0 estimation system than by differences in cochlear filter bandwidths per se. We suspect that the extremely good pitch discrimination thresholds of humans reflect some of the unique demands of speech and music, which may require humans to be more sensitive to small F0 differences than other species. By contrast, other species may not require comparably fine-grained discrimination abilities. This is consistent with larger discrimination thresholds in many non-human animals (e.g., close to an octave for ferrets).

We have expanded our discussion of this issue in the relevant section of the Discussion:

“While our results suggest that differences in cochlear filters alone are unlikely to explain absolute differences in pitch perception abilities across species, they leave open the possibility that human pitch abilities reflect the demands of speech and music, which plausibly require humans to be more sensitive to small F0 differences than other species. This issue could be clarified by optimizing network representations for different auditory tasks.” (lines 771-776)

Figure 8- while tuning gets sharper in deep layers of the network, its worth noting that it doesn't look much like neural tuning curves (which have more of a gaussian shape), and the background level of activity (relative to the peak mean activation) is the biggest factor changing. However, looking at F0 tuning to a complex tone alone may not elucidate the processing happening between layers. At the very least you should compare pure tone and missing fundamental tuning. There is evidence of this occurring potentially in example units 3642 and unit 4605 (Fig 8A), but the prevalence of this, and how early it occurs is important to show.

We agree that the F0 tuning curves we had originally provided do not make for a good comparison to neural data – we constructed them by averaging tuning curves across large numbers of units (aligning the best F0s before averaging), which produces quite different shapes than those reported in neurophysiology experiments. We overhauled this figure in response to this comment and related comments from Reviewer 2. We now show F0 tuning curves measured from representative single units in several different layers in response to stimuli with different harmonic composition (including pure tones, complex tones containing the fundamental, and missing-fundamental complex tones), which better illustrates how F0 tuning properties emerge within the model.

Overall, the results of this analysis at the early and deep layers are qualitatively consistent with previous physiological observations: the early layers exhibit tuning to frequency, showing responses to each resolved harmonic of complex tones. By contrast, the deep layers show tuning both to pure tone frequency and the F0 of missing-F0 complex tones. Intermediate layers are more difficult to interpret, but typically have broadband frequency tuning.

Here is the revised figure (Fig. 9 in the revised manuscript):

A. Example stimuli and hypothetical idealized tuning curves

B. Emergence of F0 tuning in an example network

C. Inharmonic complex tones disrupt F0 tuning

D. Population responses as a function of lowest harmonic number

networks, but only for harmonic tones. **(D)** Left: Population responses of pitch-selective units in marmoset auditory cortex, human auditory cortex, and our model's output layer, plotted as a function of lowest harmonic number. Marmoset single-unit recordings were made from 3 animals and error bars indicate SEM across 50 neurons (re-plotted from ²⁷). Center: Human fMRI responses to harmonic tones, as a function of their lowest harmonic number. Data were collected from 13 participants and error bars indicate within-subject SEM (re-plotted from ²⁸). Responses were measured from a functional region of interest defined by a contrast between harmonic tones and frequency-matched noise. Responses were measured in independent data (to avoid double dipping). Right: Network unit activations to harmonic tones as a function of lowest harmonic number. Activations were averaged across all units in the final fully-connected layer of our 10 best network architectures (error bars indicate 95% confidence intervals bootstrapped across the 10 best network architectures).

Figure 9. Network neurophysiology. Network activations were measured in response to pure tones and complex tones with four different harmonic compositions. **(A)** Left: Power spectra for stimuli with 200 Hz F0. Center: expected F0 tuning curves for an idealized frequency-tuned unit. The tuning curves are color-matched to the corresponding stimulus (e.g., black for pure tones and red for harmonics 6-14). A frequency-tuned unit should respond to pure tones near its preferred frequency (414 Hz) or to complex tones containing harmonics near its preferred frequency (e.g., when F0 = 212, 138, 103.5, or 82.8 Hz, i.e. 414/2, 414/3 or 414/4 Hz). Right: expected F0 tuning curves for an idealized F0-tuned unit. An F0-tuned unit should produce tuning curves that are robust to harmonic composition. The strength of a unit's F0 tuning can thus be quantified as the mean correlation between the pure tone (frequency) tuning curve and each of the complex tone tuning curves. **(B)** F0 tuning curves measured from five representative units in each of five network layers. Units in the first layer (relu_0) seem to exhibit frequency tuning. Units in the last layer (fc_top) exhibit F0 tuning. **(C)** Left: Nominal F0 tuning curves were measured for complex tones made inharmonic by jittering component frequencies. Center: Such curves are shown for one example unit in the network's last layer. Unlike for harmonic tones, the tuning curves for tones with different frequency compositions do not align. Right: The overall F0 tuning of a network layer was computed by averaging the F0 tuning strength across all units in the layer. A unit's F0 tuning strength was quantified as the mean correlation between the pure tone (frequency) tuning curve and each of the complex tone tuning curves. For each of our 10 best network architectures, overall F0 tuning (computed separately using either harmonic or inharmonic complex tones) is plotted as a function of network layer. Network units become progressively more F0-tuned deeper into the

And the revised Results section describing these results:

“We simulated electrophysiology experiments on our best-performing network architecture by measuring time-averaged model unit activations to pure and complex tones varying in harmonic composition (Fig. 9A). F0 tuning curves of units in different network layers (Fig. 9B) illustrate a transition from frequency-tuned units in the first layer (relu_0, where units responded whenever a harmonic of a complex tone aligned with their pure-tone tuning) to complex tuning in intermediate layers (relu_2, relu_4, and fc_int) to unambiguous F0 tuning in the final layer (fc_top), where units responded selectively to specific F0s across different harmonic compositions. These latter units thus resemble pitch-selective neurons identified in primate auditory cortex²⁷ in which tuning to the F0 of missing-fundamental complexes aligns with pure tone tuning.

We quantified the F0 tuning of individual units by measuring the correlation between pure tone and complex tone tuning curves. High correlations between tuning curves indicate F0 tuning invariant to harmonic composition. In each of the 10 best-performing networks, units became progressively more F0-tuned deeper into the network (Fig. 9C, right, solid symbols). Critically, this result depended on the harmonicity of the tones. When we repeated the analysis with complex tones made inharmonic by jittering component frequencies¹⁹ (Fig. 9C, left), network units no longer showed F0 tuning (Fig. 9B, center) and the dependence on network layer was eliminated (Fig. 9C, right, open symbols). In this respect the units exhibit a signature of human F0-based pitch, which is also disrupted by inharmonicity^{19,54}, and of pitch-tuned neurons in non-human primates⁵⁵.

To compare the population tuning to that observed in the auditory system, we also measured unit activations to harmonic complexes as a function of the lowest harmonic in the stimulus. The F0-tuned units in our model’s final layer responded more strongly when stimuli contained low-numbered harmonics (Fig. 9D, right; main effect of lowest harmonic number on mean activation, $F(1.99,17.91)=134.69$, $p<0.001$, $\eta^2_{\text{partial}} = 0.94$). This result mirrors the response characteristics of pitch-selective neurons (measured with single-unit electrophysiology) in marmoset auditory cortex (Fig. 9D, left)²⁷ and pitch-selective voxels (measured with fMRI) in human auditory cortex (Fig. 9D, center)²⁸.” (lines 557-588)

Reviewer #2 (Remarks to the Author)

Review of “Deep neural network models reveal interplay of peripheral coding and stimulus statistics in pitch perception” by Saddler, Gonzales and McDermott

This paper reports the results of what I believe is an important step forward in hearing sciences, advancing significantly our knowledge on pitch-perception auditory mechanisms in humans (pitch perception being a capacity playing a key role in speech and music recognition). More broadly, this paper also advances significantly our understanding of the complex interactions between environmental and biological constraints in perception.

The study is based on an original computational approach making elegant use of a classical model of the human cochlea (our sensory receptor for the auditory modality) together with recent advances in machine-learning techniques. This approach allows a series of experimental tests that cannot be conducted in real human listeners: i) manipulating systematically spectral and temporal coding fidelity at the peripheral (cochlear) level; ii) training the central architecture of the pitch-discrimination system with speech, music, or synthetic tones with or without altered characteristics). In short, this study makes two extremely valuable contributions for a journal like Nature Communication: not only does it provide a clear answer to a debate on the role of peripheral sensory mechanisms that has been revolving for more than half a century (ie, whether pitch perception is based on temporal vs spectral/spatial neural cues available at the sensory-receptor level), but it also provides clear answers to another fundamental issue in sensory sciences (ie, whether characteristics of natural stimuli shape central sensory mechanisms). In addition to that, this study makes surprising discoveries on intriguing aspects of pitch perception, such as the “auditory perception of the missing fundamental”, by pointing to a role of hearing in noise. I entirely agree with the authors that the current approach based on an optimization procedure is useful to provide important insights into human behaviour, because the ideal observers are quite difficult - if not impossible to figure out - for the (relatively complex) perceptual tasks tested here. The fact that the best-performing deep neural networks mimicked both qualitatively and quantitatively human data relatively well is another important theoretical finding of this study, revealing that humans make a near-optimal use of the low-level sensory cues conveyed by their cochleas.

The manuscript is extremely well written and made easy to follow for a wide audience including of course psychophysicists, neuroscientists, modellers, but also physicists, computer scientists and clinicians with some background in neurosciences and experimental audiology (eg, ENT clinicians and audiologists). All figures are well designed and present the main findings in a straightforward way. The vast literature on the present topic is well reviewed. The method is sound and the paper provides enough detail for the work to be reproduced; the data are well analysed and well discussed. For all these reasons, I believe that this manuscript deserves being published in Nature Communication. For the same reasons, this paper certainly merits being cited in future hearing-science manuals. I have only six specific but minor comments to make on this study that the authors may wish to take into account to improve their manuscript.

Thank you!

Specific comments:

1-Dependence on low-order harmonics (experiment A; cf. Fig 2A): The authors acknowledge several times that the inflection point on the curve showing simulated discrimination data was lower than in real human listeners. I suspect that this reflects characteristics of the model of peripheral auditory

system used in here. The authors should elaborate more on this relative failure of the modelling approach, and attempt to explain why the inflection moves to the 10-15th harmonic in real listeners.

The difference in transition point may reflect shortcomings of the peripheral auditory model, but we do not know what those might be. We tried manipulating both the frequency tuning (broader and narrower filters) and spike-timing information (increasing and decreasing the phase-locking cutoff), and neither of these produced a better quantitative fit. In response to Reviewer #3's comments, we also tried replacing the hard-coded peripheral auditory model with a layer of learnable filters operating directly on the audio. We note that the transition point of these networks remains lower than those reported for humans.

We suspect the discrepancy is therefore more likely to be a shortcoming of the training dataset (which we demonstrate in Figure 6 can have a drastic effect on the transition point). We chose speech and music with F0s ranging from 80 to 1000 Hz for our dataset because (1) we felt this was a reasonable (but of course not exhaustive) representation of the sounds for which pitch is important to humans and (2) speech and music audio data covering this range of F0s was readily available in the quantities required for supervised machine learning. However, it seems likely that the distribution of our training data is not exactly matched to the sound distribution that constrains human pitch perception. In addition, the task our models are trained on is not exactly matched to the set of pitch-related tasks humans might be optimized for. It is thus probably not surprising to get some quantitative discrepancy between the human and model results.

We have added a brief discussion of these points to the revised manuscript:

“For instance, the inflection point in the graph of Fig. 2A occurs at a somewhat lower harmonic number in the model than in humans. Given the evidence presented here that pitch perception reflects the stimulus statistics a system is optimized for, some discrepancies might be expected from the training set, which (due to the limitations of available corpora) consisted entirely of speech and musical instrument sounds, and omitted other types of natural sounds that are periodic in time. The range of F0s we trained on was similarly limited by available audio data sets, and prevents us from making predictions about the perception of very high frequencies⁵⁸. The uniform distributions over sound level and SNR in our training dataset were also not matched in a principled way to the natural world. Discrepancies may also reflect shortcomings of our F0 estimation task (which used only 50ms clips) or peripheral model, which although state-of-the-art and relatively well validated, is imperfect (e.g., peripheral representations consisted of firing rates rather than spikes).” (lines 694-706)

2-Perception of transposed tones (experiment E; cf. Fig 2E; Figure 3E): this now “classical” result of the literature suggesting that a correct spatial (tonotopic representation) is requested for normal pitch perception is the only situation where the accuracy of model predictions are not related to the optimization process. Here again, the authors should elaborate on the reasons why this task does not follow the same trends as the other tasks.

Our results suggest that the transposed tones psychophysical result is simply a very robust effect that virtually any model trained to estimate F0 from natural sounds will exhibit. Our interpretation is that a system does not need to be highly optimized in order

to replicate this effect, because the transposed tones present patterns of stimulation that never occur for sounds with that F0. Thus, essentially any model that learns to associate naturally occurring peripheral cues with F0 will exhibit poor performance for transposed tones.

We have clarified this issue in the revised paper:

“For four of the five experiments (Fig. 3A-D), there was a significant positive correlation between training task performance and human-model similarity ($p < 0.001$ in each case). The transposed tones experiment (Fig. 3E) was the exception, as all networks similarly replicated the main human result regardless of their training task performance. We suspect this is because transposed tones cause patterns of peripheral stimulation that rarely occur for natural sounds. Thus, virtually any model that learns to associate naturally-occurring peripheral cues with F0 (regardless of how well it is optimized) will exhibit poor performance for transposed tones.” (lines 245-252)

3-I was amazed by the relatively modest effects of altering cochlear frequency selectivity on the model's performance. The results are convincing and this clearly is a major finding of the present study as most members of the hearing science community would assume that cochlear frequency selectivity strongly constrains pitch discrimination. As to the demonstration of the role of neural phase locking (ie, temporal fine structure) cues in pitch perception, I was especially fascinated by the fact that an upper limit of neural phase locking at 3kHz and above is required to reproduce the invariance of pitch perception to level shown by real human listeners. This as such is another key finding of the present study, highlighting the reason why neural phase locking is required to obtain robust perception for normal-hearing listeners. This finding has also important audiological implications for the understanding of the detrimental effects of cochlear damage (eg, caused by ageing or noise exposure) on auditory perception for hearing-impaired people.

Thank you.

Recent neurophysiological studies indicate that cochlear damage causes a mismatch between neural temporal-fine structure information and the place on the basilar membrane that would “normally” respond to that information (Henry et al., 2013, 2016, 2019). I guess that this impact of such cochlear alterations on pitch perception may be simulated with the present approach by modifying certain aspects of the cochlear model (eg, by modifying the tip/tail component of cochlear-filters tuning curves). Such data would certainly trigger a huge interest in clinicians and hearing-aid manufacturers.

The coh parameter in the Bruce et al. (2018) auditory nerve model controls the contribution of the OHC and is a convenient way to model some effects of hearing loss (i.e., reducing the tip component of the cochlear filter tuning curves). However, in our experiments with the model we have always found the best frequency to remain unchanged when the simulated OHC contribution is reduced, in contrast to the published results reporting shifts in best frequency that would induce the mismatches described by the reviewer. Here are some example tuning curves from the model:

Tuning curves with OHC loss

We thus feel that modeling these effects properly will require customizing the nerve model. This is a serious undertaking that is beyond the scope of our paper.

More generally, we think the application of these methods to hearing impairment is an important direction in its own right, and deserves a dedicated paper if done properly (this is very much in our future plans). We have thus noted that this is a potentially impactful direction for the near future:

“The approach we used here has natural extensions to understanding other aspects of hearing⁶⁶, in which similar questions about the roles of peripheral cues have remained unresolved. Our methods could also be extended to investigate hearing impairment, which can be simulated with alterations to standard models of the cochlea⁶⁷ and which often entails particular types of deficits in pitch perception⁵⁰. Prostheses such as cochlear implants are another natural application of task-optimized modeling. Current implants restore some aspects of hearing relatively well, but pitch perception is not one of them⁶⁸. Models optimized with different types of simulated electrical stimulation could clarify the patterns of behavior to expect. Models trained with either acoustically- or electrically-stimulated peripheral auditory representations (or combinations thereof) and then tested with electrically-stimulated input could yield insights into the variable outcomes of pediatric cochlear implantation. Similar approaches could be applied to study acclimatization to hearing aids in adults.” (lines 754-766)

4-Ability to extract F0 information from high-number harmonics and perception of missing fundamental: A very intriguing result suggests that this ability results partly from exposure to noise during training (eg, during auditory development). Some elaborations, or even speculations, would be welcome here because noise is expected to corrupt any kind of (temporal and/or spectral) cues rather than helping the discrimination process. One may speculate about a potential “filling in” phenomenon caused by noise. For instance, could it be the case that background noise elicits (poor) temporal cues at the output of low center-frequency (CF) cochlear channels tuned at or close the (missing) F0 that may assist the optimization process? In other words, at low CF channels required during the training phase (to test this, these channels may be turned off)?

We think that the absence of noise makes the most obvious cue to F0 (the frequency of the lowest harmonic) sufficient for solving the task, and the networks learn to exploit this. The presence of noise, which often obscures the lowest harmonic, makes the network adopt a different strategy (i.e., one that does not just latch on to the lowest harmonic) and when this strategy is tested on the psychophysical stimuli, the result is better discrimination of tones with exclusively higher harmonics. We note that this general result is consistent with other findings that machine learning models are more

likely to learn a robust and general strategy when noise or other forms of uncertainty are added to the training set.

We have clarified this issue in the revised discussion section:

“Networks trained without background noise did not extract F0 information from high-numbered harmonics, relying instead solely on the lowest-numbered harmonics. Such a strategy evidently works well for idealized environments (where the lowest harmonics are never masked by noise), but not for realistic environments containing noise, and diverges from the strategy employed by human listeners. This result suggests that pitch is also in part a consequence of needing to hear in noise.” (lines 660-666)

5-Network physiology: Simulated electrophysiology experiments on the best-performing networks showed that units in the deepest layers were selectively and sharply tuned to F0, consistent with electrophysiological data for non-human animals and brain-imaging data for humans. This is a nice result that certainly deserves further characterization. How does such a tuning emerge in the processing architecture? What are the low-level mechanisms allowing for the conversion of peripheral (time, place, time and place) information into a “rate-place” code for pitch?

We completely re-worked Fig. 8 to better illustrate what is going on in our model. We show pure and complex tone F0 tuning curves for representative units at several different layers. They illustrate a transition from somewhat interpretable frequency-tuned units in the earliest layer, to complex and difficult-to-describe tuning curves in intermediate layers, to unambiguous F0 tuning in the final layer (i.e., similar tuning to pure tones and missing-fundamental complex tones).

Here is the revised figure (Fig. 9 in the revised manuscript):

A. Example stimuli and hypothetical idealized tuning curves

B. Emergence of F0 tuning in an example network

C. Inharmonic complex tones disrupt F0 tuning

D. Population responses as a function of lowest harmonic number

networks, but only for harmonic tones. (D) Left: Population responses of pitch-selective units in marmoset auditory cortex, human auditory cortex, and our model's output layer, plotted as a function of lowest harmonic number. Marmoset single-unit recordings were made from 3 animals and error bars indicate SEM across 50 neurons (re-plotted from ²⁷). Center: Human fMRI responses to harmonic tones, as a function of their lowest harmonic number. Data were collected from 13 participants and error bars indicate within-subject SEM (re-plotted from ²⁸). Responses were measured from a functional region of interest defined by a contrast between harmonic tones and frequency-matched noise. Responses were measured in independent data (to avoid double dipping). Right: Network unit activations to harmonic tones as a function of lowest harmonic number. Activations were averaged across all units in the final fully-connected layer of our 10 best network architectures (error bars indicate 95% confidence intervals bootstrapped across the 10 best network architectures).

Figure 9. Network neurophysiology. Network activations were measured in response to pure tones and complex tones with four different harmonic compositions. (A) Left: Power spectra for stimuli with 200 Hz F0. Center: expected F0 tuning curves for an idealized frequency-tuned unit. The tuning curves are color-matched to the corresponding stimulus (e.g., black for pure tones and red for harmonics 6-14). A frequency-tuned unit should respond to pure tones near its preferred frequency (414 Hz) or to complex tones containing harmonics near its preferred frequency (e.g., when F0 = 212, 138, 103.5, or 82.8 Hz, i.e. 414/2, 414/3 or 414/4 Hz). Right: expected F0 tuning curves for an idealized F0-tuned unit. An F0-tuned unit should produce tuning curves that are robust to harmonic composition. The strength of a unit's F0 tuning can thus be quantified as the mean correlation between the pure tone (frequency) tuning curve and each of the complex tone tuning curves. (B) F0 tuning curves measured from five representative units in each of five network layers. Units in the first layer (relu_0) seem to exhibit frequency tuning. Units in the last layer (fc_top) exhibit F0 tuning. (C) Left: Nominal F0 tuning curves were measured for complex tones made inharmonic by jittering component frequencies. Center: Such curves are shown for one example unit in the network's last layer. Unlike for harmonic tones, the tuning curves for tones with different frequency compositions do not align. Right: The overall F0 tuning of a network layer was computed by averaging the F0 tuning strength across all units in the layer. A unit's F0 tuning strength was quantified as the mean correlation between the pure tone (frequency) tuning curve and each of the complex tone tuning curves. For each of our 10 best network architectures, overall F0 tuning (computed separately using either harmonic or inharmonic complex tones) is plotted as a function of network layer. Network units become progressively more F0-tuned deeper into the

and here is the updated section of the Results:

“We simulated electrophysiology experiments on our best-performing network architecture by measuring time-averaged model unit activations to pure and complex tones varying in harmonic composition (Fig. 9A). F0 tuning curves of units in different network layers (Fig. 9B) illustrate a transition from frequency-tuned units in the first layer (relu_0, where units responded whenever a harmonic of a complex tone aligned with their pure-tone tuning) to complex tuning in intermediate layers (relu_2, relu_4, and fc_int) to unambiguous F0 tuning in the final layer (fc_top), where units responded selectively to specific F0s across different harmonic compositions. These latter units thus resemble pitch-selective neurons identified in primate auditory cortex²⁷ in which tuning to the F0 of missing-fundamental complexes aligns with pure tone tuning.

We quantified the F0 tuning of individual units by measuring the correlation between pure tone and complex tone tuning curves. High correlations between tuning curves indicate F0 tuning invariant to harmonic composition. In each of the 10 best-performing networks, units became progressively more F0-tuned deeper into the network (Fig. 9C, right, solid symbols). Critically, this result depended on the harmonicity of the tones. When we repeated the analysis with complex tones made inharmonic by jittering component frequencies¹⁹ (Fig. 9C, left), network units no longer showed F0 tuning (Fig. 9B, center) and the dependence on network layer was eliminated (Fig. 9C, right, open symbols). In this respect the units exhibit a signature of human F0-based pitch, which is also disrupted by inharmonicity^{19,54}, and of pitch-tuned neurons in non-human primates⁵⁵.

To compare the population tuning to that observed in the auditory system, we also measured unit activations to harmonic complexes as a function of the lowest harmonic in the stimulus. The F0-tuned units in our model’s final layer responded more strongly when stimuli contained low-numbered harmonics (Fig. 9D, right; main effect of lowest harmonic number on mean activation, $F(1.99,17.91)=134.69$, $p<0.001$, $\eta^2_{\text{partial}} = 0.94$). This result mirrors the response characteristics of pitch-selective neurons (measured with single-unit electrophysiology) in marmoset auditory cortex (Fig. 9D, left)²⁷ and pitch-selective voxels (measured with fMRI) in human auditory cortex (Fig. 9D, center)²⁸.” (lines 557-588)

6-Future directions: By showing that pitch discrimination depends both on the information available at the output of the peripheral auditory system and on the environmental constraints in which the auditory system is optimized, this study opens an exciting path that may renew the study of auditory development (with and without sensorineural hearing loss; eg understanding the variable outcome of paediatric cochlear implantation) or auditory acclimatization to hearing aids for adults. The authors may wish to make a short note pointing in this direction (this is already suggested on line 653).

Thanks for pointing this out. We agree and added a short note in the discussion:

“Prostheses such as cochlear implants are another natural application of task-optimized modeling. Current implants restore some aspects of hearing relatively well, but pitch perception is not one of them⁶⁸. Models optimized with different types of simulated electrical stimulation could clarify the patterns of behavior to expect. Models trained with either acoustically- or electrically-stimulated peripheral auditory representations (or combinations thereof) and then tested with electrically-stimulated

input could yield insights into the variable outcomes of pediatric cochlear implantation. Similar approaches could be applied to study acclimatization to hearing aids in adults.”
(lines 758-766)

References:

Heinz MG, Henry KS (2013) Modeling disrupted tonotopicity of temporal coding following sensorineural hearing loss. Proc Mtgs Acoust 19:050177.

Henry KS, Kale S, Heinz MG (2016) Distorted tonotopic coding of temporal envelope and fine structure with noise-induced hearing loss. J Neurosci 36:2227–2237.

Henry KS, Sayles M, Hickox, ME, HeinzMG (2019) Divergent Auditory Nerve Encoding Deficits Between Two Common Etiologies of Sensorineural Hearing Loss. J Neurosci, 39(35):6879–6887

Reviewer #3 (Remarks to the Author)

Review of “Deep neural network models reveal interplay of peripheral coding and stimulus statistics in pitch perception” by Saddler et al.

This article proposes a computational approach towards teasing out the components of pitch perception attributable to ear physiology as opposed to the environment. The authors use an approach that has become common in computational neuroscience, namely to train deep neural networks for a task, pitch discrimination in this case, and to utilize the network to gain normative insights into how the brain achieves the task. The authors trained deep neural networks for pitch estimation using natural sounds pre-processed with a cochlear model and found that the networks exhibit human-like behavior. The authors demonstrate that altering the statistics of the sound used to train the networks alters the behavior of the networks, making them less human-like for “un-natural” sounds, which, they suggest, points to an environmental component to pitch perception. The novelty of this work seems to stem, not on the use of deep learning per se, but on its use in conjunction with inputs of different statistics to arrive at the conclusion of an environmental component to pitch perception. I appreciate the thoroughness with which the authors describe their deep learning experiments, and the experiments performed to compare the performance of ‘the model’ to humans. I found the fact that ‘the model’ exhibits human-like behavior interesting, as as the authors’ characterization of network psychophysics.

Thank you!

Below, I make a few comments/suggestions to the authors that I believe can help to improve the manuscript and, possibly, lead to interesting computational findings.

1 Can the architecture learn the cochlear pre-processing module from data?

By using a cochlear model as a pre-processing, I understand the authors’ motivation to impose physiological constraints on the architecture. Personally, I would find the authors’ finding much more interesting if the networks could learn this pre-processing stage. As far as I understand, from a deep learning perspective, this stage has a simple interpretation that one could specify as trainable layer. In the section “Human-like behavior is less dependent on cochlear filter bandwidths”, the authors say that “Linearly-spaced cochlear filters also yielded best thresholds that were not significantly different from those for normal human tuning...”. The Fourier basis represents the simplest example of a linearly-spaced filter bank (you can think of the filters as either complex exponentials or sines/cosines). In the time domain, we can interpret filtering operations as convolutions, suggesting that the authors can replace the cochlear pre-processing model with a convolutional layer with one-dimensional convolutional filters (100 of them for the authors), followed by a Fourier transform and a magnitude nonlinearity to obtain. Applying this in successive windows of input sounds would yield a 100 x 1000 representation of the same shape as that obtained by the cochlear module. I would suggest to the authors that they train the filters from this layer. In my opinion, analyzing the filters from this layer and comparing them to cochlear processing could reveal differences that lead to (likely) better pitch discrimination than with the cochlear module. I also suspect that training this layer for sound from different statistics may close the gap the authors see between speech and music performance. In other words, I suspect that the difference the authors see for different sound statistics, which I personally do not find surprising, comes from the cochlear-processing step. I would find it interesting that the architecture learns a filter bank tailored to statistics of the input sounds. Note: What I suggest may involve neural nets with complex values, depending on implementation. There exists a rich literature on the topic.

We performed the suggested experiment by replacing the hardwired peripheral auditory model with a bank of 100 learnable cochlear filters operating directly on the audio. The results were interesting. Networks optimized directly on the sound waveforms exhibit qualitatively different (and less human-like) psychophysical behavior than networks optimized on the simulated nerve representations. Networks whose “cochlear representations” (first layer filters) were optimized specifically for F0 estimation tended to rely less on high-numbered harmonics. Performance with high-numbered harmonics was very poor and effects of harmonic phase were not evident. This result provides some support for the approach we adopted, in which we hard-wired the cochlear model. The most obvious interpretation is that the cochlea is constrained by multiple tasks or functions, and that its form influences the solution to f0 estimation arrived at by the brain (and our model). We present these findings in a new figure (Fig. 4 in the revised manuscript).

Figure 4. Networks trained to estimate F0 directly from sound waveforms exhibit less human-like pitch behavior. (A) Schematic of model structure. Model architecture was identical to that depicted in Fig. 1A, except that the hardwired cochlear input representation was replaced by a layer of 1-dimensional convolutional filters operating directly on sound waveforms. The first-layer filter kernels were optimized for the F0 estimation task along with the rest of the network weights. We trained the 10 best networks from our architecture search with learnable first-layer filters. (B) The best frequencies (sorted from lowest to highest) of the 100 learned filters for each of the 10 network architectures are plotted in magenta. For comparison, the best frequencies of the 100 cochlear filters in the hardwired peripheral model are plotted in black. (C) Effect of learned cochlear filters on network behavior in all five main psychophysical experiments (see Fig. 2A-E): F0 discrimination as a function of harmonic number and phase (Expt. A), pitch estimation of alternating-phase stimuli (Expt. B), pitch estimation of frequency-shifted complexes (Expt. C), pitch estimation of complexes with individually mistuned harmonics (Expt. D), and frequency discrimination with pure and transposed tones (Expt. E). (D) Comparison of human-model similarity metrics between networks trained with either the hardwired cochlear model (black) or the learned cochlear filters (magenta) for each psychophysical experiment. Asterisks indicate significance of two-sample t-tests

comparing the two cochlear model conditions: *** $p < 0.001$, * $p < 0.05$. Human-model similarity scores were significantly lower for networks with learned cochlear filters in Expt. A and Expt. B. Error bars indicate 95% confidence intervals bootstrapped across the 10 network architectures.

Here is the new section of the Results text that describes this new experiment:

“To first determine whether a biologically-constrained cochlear model was necessary for human-like pitch behavior, we trained networks to estimate F0 directly from sound waveforms (Fig. 4A). We replaced the cochlear model with a bank of 100 one-dimensional convolutional filters operating directly on the audio. The weights of these first-layer filters were optimized for the F0 estimation task along with the rest of the network.

The learned filters deviated from those in the ear, with best frequencies tending to be lower than those of the hardwired peripheral model (Fig. 4B). Networks with learned cochlear filters also exhibited less human-like behavior than their counterparts with the fixed cochlear model (Fig. 4C&D). In particular, networks with learned cochlear filters showed little ability to extract pitch information from high-numbered harmonics. Discrimination thresholds for higher harmonics were poor (Fig. 4C, Expt. A) and networks did not exhibit phase effects (Fig. 4C, Expt. A & B). Accordingly, human-model similarity was significantly lower with learned cochlear filters for two of five psychophysical experiments (Fig. 4D; Expt. A: $t(18)= 5.23$, $p<0.001$, $d=2.47$; Expt. B: $t(18)=12.69$, $p<0.001$, $d=5.98$). This result suggests that a human-like cochlear representation is necessary to obtain human-like behavior, but also that the F0 estimation task on its own is insufficient to produce a human-like cochlear representation, likely because the cochlea is shaped by many auditory tasks. Thus, the cochlea may be best considered as a constraint on pitch perception rather than the other way around.” (lines 288-309)

We also trained the network with learnable cochlear filters exclusively on speech or music stimuli. The discrimination thresholds for the two resulting models were more similar than when the cochlear filters were fixed, but the best thresholds for music-only-trained networks remained significantly lower than those for speech-only-trained networks. We believe this difference reflects the fact that the instrument sounds are more similar to the synthetic test stimuli than are the speech sounds (in which the F0 is constantly fluctuating). Nonetheless, this was interesting experiment to run and we incorporated it as a new supplemental figure (Supplemental Fig. 9 in the revised manuscript):

Supplemental Figure 9. F0 discrimination thresholds as a function of lowest harmonic number, measured from networks trained separately on speech-only and music-only datasets. **(A)** Results from networks trained on simulated auditory nerve representations produced by a fixed peripheral auditory model (reproduced from Fig. 7C). **(B)** Results from networks trained directly on sound waveforms (first-layer “cochlear” filters are learned alongside the rest of the network weights; see Fig. 4). Error bars indicate 95% confidence intervals bootstrapped across the 10 best network architectures.

2 Network averaging and Bayesian deep learning/ensemble models

By averaging networks to obtain ‘the model’, the authors essentially form deep ensembles [1]. These have garnered recent interest, particularly in the context of Bayesian deep learning [2]. I would encourage the authors to cite this literature, put ‘the model’ explicitly in the context of this literature, and to think about ways in which they currently/could leverage Bayesian deep learning (perhaps without knowing it) in their work. In my opinion, doing so could help to raise awareness, within the neuroscience community, as to the issues of uncertainty quantification in deep learning.

Thank you for the suggestion. We have included citations and a brief discussion of this literature in the revised paper:

“Given evidence for individual differences across different networks optimized for the same task⁴⁴, most figures feature results averaged across the 10 best networks identified in our architecture search (which we collectively refer to as ‘the model’). Averaging across an ensemble of networks effectively allows us to marginalize over architectural hyperparameters and provide uncertainty estimates for our model’s results^{45,46}.” (lines 173-178)

References

[1] Lakshminarayanan, B., Pritzel, A., & Blundell, C. “Simple and scalable predictive uncertainty estimation using deep ensembles”. In *Advances in Neural Information Processing Systems*, pp. 6402-6413, 2017.

[2] Wilson, Andrew Gordon. “The case for Bayesian deep learning.” arXiv preprint arXiv:2001.10995 (2020).

REVIEWERS' COMMENTS

Reviewer #1 (Remarks to the Author):

the revised manuscript looks great. No further comments.

Reviewer #2 (Remarks to the Author):

The authors answered all my questions, and included these elements in the revised version of their manuscript. I am quite satisfied with this new version and have no further comments to make. My congratulations to the authors. This study should have a substantial impact on the field of hearing sciences.

Reviewer #3 (Remarks to the Author):

I thank the authors for addressing my comments related to the deep-learning aspects of the article. I recommend the manuscript for acceptance.

Demba Ba.

Saddler et al. Response to Reviews

Reviewer #1 (Remarks to the Author): the revised manuscript looks great. No further comments.

Thank you!

Reviewer #2 (Remarks to the Author):

The authors answered all my questions, and included these elements in the revised version of their manuscript. I am quite satisfied with this new version and have no further comments to make. My congratulations to the authors. This study should have a substantial impact on the field of hearing sciences.

Thank you!

Reviewer #3 (Remarks to the Author):

I thank the authors for addressing my comments related to the deep-learning aspects of the article. I recommend the manuscript for acceptance.

Demba Ba.

Thank you!